# The αC-β4 loop controls the allosteric cooperativity between nucleotide and substrate in the catalytic subunit of protein kinase A

Cristina Olivieri[1†‡], Yingjie Wang[1,2†§], Caitlin Walker[1], Manu Veliparambil Subrahmanian[1], Kim N Ha[3], David Bernlohr[1], Jiali Gao[2], Carlo Camilloni[4#], Michele Vendruscolo[4], Susan S Taylor[5,6], Gianluigi Veglia[1,2*]

[1]Department of Biochemistry, Molecular Biology, and Biophysics, University of Minnesota, Minneapolis, United States; [2]Department of Chemistry and Supercomputing Institute, University of Minnesota, Minneapolis, United States; [3]Department of Chemistry and Biochemistry, St. Catherine University, Minneapolis, United States; [4]Department of Chemistry, University of Cambridge, Cambridge, United Kingdom; [5]Department of Pharmacology, University of California at San Diego, San Diego, United States; [6]Department of Chemistry and Biochemistry, University of California at San Diego, San Diego, United States

**\*For correspondence:**
vegli001@umn.edu

[†]These authors contributed equally to this work

**Present address:** [‡]Department of Chemistry, University of Milan, Milan, Italy; [§]Institute of Systems and Physical Biology, Shenzhen Bay Laboratory, Shenzhen, China; [#]Department of Bioscience, University of Milano, Milano, Italy

**Competing interest:** The authors declare that no competing interests exist.

**Abstract** Allosteric cooperativity between ATP and substrates is a prominent characteristic of the cAMP-dependent catalytic subunit of protein kinase A (PKA-C). This long-range synergistic action is involved in substrate recognition and fidelity, and it may also regulate PKA's association with regulatory subunits and other binding partners. To date, a complete understanding of this intramolecular mechanism is still lacking. Here, we integrated NMR(Nuclear Magnetic Resonance)-restrained molecular dynamics simulations and a Markov State Model to characterize the free energy landscape and conformational transitions of PKA-C. We found that the apoenzyme populates a broad free energy basin featuring a conformational ensemble of the active state of PKA-C (ground state) and other basins with lower populations (excited states). The first excited state corresponds to a previously characterized inactive state of PKA-C with the αC helix swinging outward. The second excited state displays a disrupted hydrophobic packing around the regulatory (R) spine, with a flipped configuration of the F100 and F102 residues at the αC-β4 loop. We validated the second excited state by analyzing the F100A mutant of PKA-C, assessing its structural response to ATP and substrate binding. While PKA-C[F100A] preserves its catalytic efficiency with Kemptide, this mutation rearranges the αC-β4 loop conformation, interrupting the coupling of the two lobes and abolishing the allosteric binding cooperativity. The highly conserved αC-β4 loop emerges as a pivotal element to control the synergistic binding of nucleotide and substrate, explaining how mutations or insertions near or within this motif affect the function and drug sensitivity in homologous kinases.

## eLife assessment

This **important** study provides an example of integrating computational and experimental approaches that lead to new insights into the energy landscape of a model kinase. **Compelling** use of molecular dynamics simulations and NMR spectroscopy provide a conformational description of active and excited states of the kinase; one of which has not been captured in previously solved

crystal structures. Overall, this comprehensive study expands our understanding of the architecture and allosteric features of the conserved bilobal kinase domain structure.

## Introduction

Eukaryotic protein kinases (EPKs) are plastic enzymes of paramount importance in signaling processes, catalyzing phosphoryl transfer reactions, or acting as scaffolds for other enzymes and/or binding partners (**Manning et al., 2002**; **Huse and Kuriyan, 2002**). Of all kinases, the catalytic subunit of protein kinase A (PKA-C) was the first to be structurally characterized by X-ray crystallography (**Knighton et al., 1991b**; **Knighton et al., 1991c**). In its inhibited state, PKA-C assembles into a heterotetrametric holoenzyme comprising two catalytic (C) and two regulatory (R) subunits (**Taylor et al., 2012**). The canonical activation mechanism of PKA involves the binding of two cAMP molecules that disassemble the holoenzyme, unleashing active PKA-C monomers, which target signaling partners (**Walsh et al., 1968**). In 2017, however, Scott and coworkers proposed an alternative activation mechanism in which the holoenzyme does not disassemble under physiological conditions; rather, it is clustered in *signaling islands*, localized near the substrates by A-kinase anchoring proteins (AKAPs), facilitating targeted signaling interactions (**Smith et al., 2017**). To date, the activation mechanism of PKA is still under active investigation since PKA-C appears to be fully liberated under pathological conditions (**Tillo et al., 2017**; **Xiong et al., 2023**; **Omar et al., 2022**).

X-ray crystallography studies revealed that PKA-C is a bilobal enzyme, with a dynamic N-terminal lobe (N-lobe) comprising four β-sheets and an αC helix, while the C-terminal lobe (C-lobe) is more rigid and mostly composed of α-helices (**Figure 1A**; **Knighton et al., 1991b**; **Knighton et al., 1991c**; **Knighton et al., 1991a**). The N-lobe harbors the nucleotide-binding site, whereas the substrate-binding cleft lays at the interface between the N- and C-lobe. The three-dimensional structure of PKA-C features a highly conserved hydrophobic core decorated with catalytically important motifs, that is, the Gly-rich loop, DFG-loop, activation loop, and magnesium and peptide positioning loops (**Johnson et al., 2001**). In the catalytically active state, these motifs are all poised for phosphoryl transfer (**Bastidas et al., 2013**; **Gerlits et al., 2015**; **Gerlits et al., 2019**), a condition necessary but not sufficient to define an active kinase. More recent studies revealed a critical role of the enzyme's hydrophobic core, which is crossed by a catalytic (C) spine and a regulatory (R) spine surrounded by

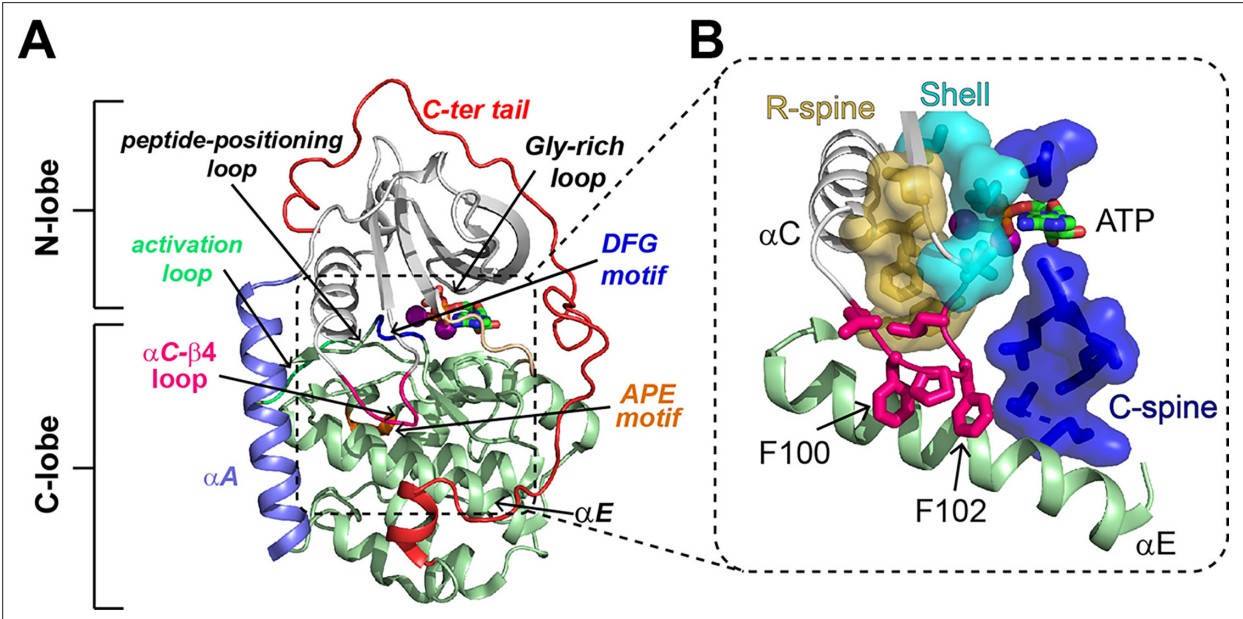

**Figure 1.** Structural and catalytic motifs of PKA-C. (**A**) Backbone representation of the ternary complex of PKA-C bound to ATP and the PKI$_{5-24}$ peptide (not depicted), PDB code 4WB5. Highlighted are key motifs including the αA, αC, and αE helices, C-terminal tail, activation loop, peptide-positioning loop, Gly-rich loop, and the DFG and APE motifs. (**B**) The hydrophobic core of PKA-C features the regulatory spine (R-spine, gold), catalytic spine (C-spine, blue), shell residues (cyan), and the αC-β4 loop (hot pink), which locks the αE helix and couples the two lobes of PKA-C.

shell residues (*Figure 1B*; *Kornev et al., 2006*; *Ten Eyck et al., 2008*). The C spine comprises an array of hydrophobic residues and is assembled upon binding ATP, whereas the R spine is engaged when the activation loop is phosphorylated at Thr197, which positions the αC-helix in an active configuration (*Hu et al., 2015*).

A distinct property of PKA-C is the binding cooperativity between ATP and substrate (*Lew et al., 1997*). In the catalytic cycle, the kinase binds ATP and unphosphorylated substrates with positive binding cooperativity, whereas a negative binding cooperativity between ADP and phosphorylated substrate characterizes the exit complex (*Wang et al., 2019*). The biological importance of these phenomena has been emphasized by our recent studies on disease-driven mutations, (*Olivieri et al., 2021*; *Walker et al., 2019*; *Walker et al., 2021*) which all feature disrupted cooperativity between nucleotides and protein kinase inhibitor (PKI) or canonical substrates (*Olivieri et al., 2021*; *Walker et al., 2019*; *Walker et al., 2021*). Since the recognition sequence of substrates is highly homologous to that of the R subunits (*Taylor et al., 2012*), a loss of cooperativity may affect not only substrate-binding fidelity but also regulatory processes. Thus far, the molecular determinants for the binding cooperativity between nucleotide and substrate and its role in PKA signalosome remain elusive.

Aiming to define the allosteric mechanism for PKA-C cooperativity, we combined NMR-restrained replica-averaged metadynamics (RAM) (*Camilloni et al., 2013*; *Camilloni and Vendruscolo, 2014*) and Markov State Model (MSM) (*Husic and Pande, 2018*) and charted the conformational landscape and dynamic transitions of the main forms of PKA-C. We found that the apo kinase occupies three distinct basins: (1) a most populated ground state (GS) with constitutively active conformations competent for catalysis, (2) a first high free energy basin representative of typical inactive states with a dislodged configuration of theαC helix (ES1), and (3) a second high free energy basin with a disrupted hydrophobic array of residues at the core of the enzyme (ES2). Notably, the equilibrium between the most populated GS and the other low-populated states agrees with previous NMR Carr–Purcell–Meiboom–Gill (CPMG) relaxation dispersion and chemical exchange saturation transfer (CEST) measurements (*Olivieri et al., 2022*). To compound the existence of the second inactive basin, we mutated F100 at the αC-β4 loop into Ala (PKA-C^F100A) and characterized its ligand-binding thermodynamics and structural response by isothermal titration calorimetry (ITC) and NMR spectroscopy, respectively. We found that PKA-C^F100A phosphorylates a canonical peptide with a catalytic efficiency similar to the wild-type kinase (PKA-C^WT); however, this mutation abolishes the binding cooperativity between nucleotide and substrate. The NMR experiments reveal that F100A perturbs the hydrophobic packing around the αC-β4 loop and interrupts the allosteric communication between the two lobes of the enzyme. Overall, these results emphasize the pivotal role of the αC-β4 loop in kinase function and may explain why single-site mutations or insertion mutations stabilizing this motif in homologous kinases result in oncogenes and confer differential drug sensitivity (*Ruan and Kannan, 2018*; *Kannan et al., 2008*; *Zhao et al., 2020*).

## Results

### The free energy landscape of PKA-C charted by NMR-restrained RAM simulations

To characterize the experimentally accessible conformational landscape of PKA-C in the μs-to-ms timescale, we performed NMR-restrained metadynamics simulations within the RAM framework (*Camilloni et al., 2013*; *Camilloni and Vendruscolo, 2014*). Using four replicas restrained with backbone chemical shifts (CS), we simulated the apo, binary (PKA-C/ATP), and ternary (PKA-C/ATP/PKI_{5-24}) complexes of PKA-C. Incorporating the CS restrains in the simulations ensures a close agreement with the conformational space explored by the kinase under our NMR experimental conditions (*Robustelli et al., 2010*; *Qi et al., 2022*), whereas the enhanced sampling through metadynamics simulations boosts the conformational plasticity of the enzyme along different degrees of freedom (i.e., collective variables, CVs, see Methods) (*Figure 2—figure supplement 1*; *Piana and Laio, 2007*). In our case, back-calculations of the CS values with Sparta+ (*Shen and Bax, 2010*) show that the restrained simulations improved the agreement between theoretical and experimental CS values of ~0.2 ppm for the amide N atoms and ~0.1 ppm for the remainder backbone atoms (*Figure 2—figure supplement 2*). Also, the bias-exchange metadynamics allow each replica to span a significantly broader conformational space relative to classical MD(Molecular Dynamics) simulations (*Figure 2—figure supplement*

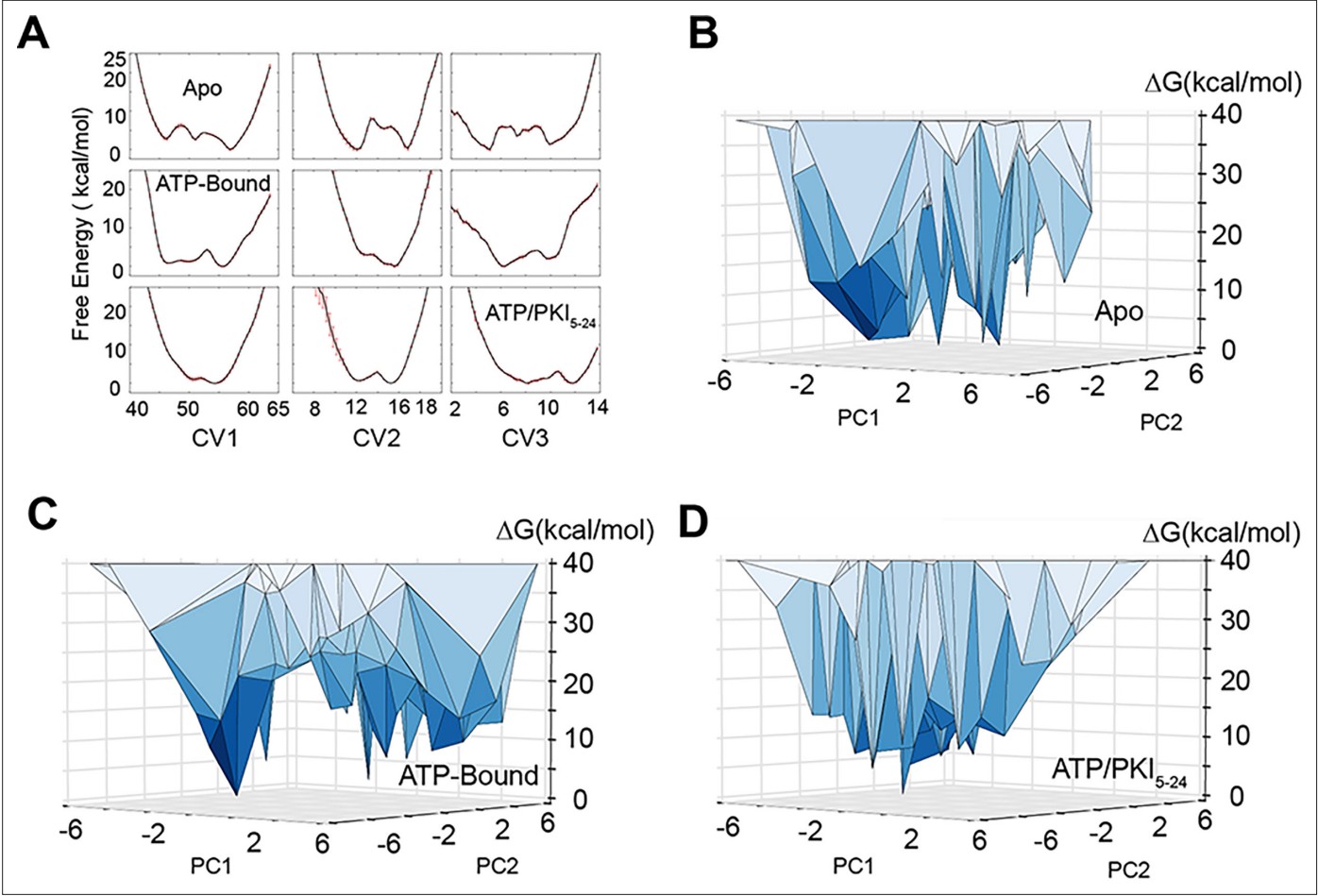

**Figure 2.** Free energy landscape of PKA-C obtained from replica-averaged metadynamics (RAM) simulations. (**A**) Convergence of the bias deposition along the first three collective variables (CVs). The free energy (expressed in kcal/mol) of the different CVs was averaged over the last 100 ns of RAM simulations. The standard deviations are reported as red error bars. (**B–D**) Free energy landscape along the first two principal components (PC1 and PC2) of PKA-C in the apo, ATP-, and ATP/PKI-bound forms. PC1 and PC2 are projected from the first three CVs. The vertices represent conformational states. In the apo form, multiple states have comparable free energy with $\Delta G < 5$ kcal/mol, whereas in the binary form, fewer states have $\Delta G < 5$ kcal/mol. For the ternary form only a major ground state is populated.

The online version of this article includes the following source data and figure supplement(s) for figure 2:

**Source data 1.** $\Delta G$ (kcal/mol) and relative population of the ground state and the first six excited states in different forms of PKA-C obtained from the replica-averaged metadynamics (RAM) simulations.

**Figure supplement 1.** Illustration of the collective variables (CVs) used in the replica-averaged metadynamics (RAM) simulations.

**Figure supplement 2.** Distribution of the root-mean-square-error (RMSE) of the chemical shifts (CSs) for the different simulation schemes.

**Figure supplement 3.** Replica-averaged metadynamics (RAM) simulations explore a larger conformational space than standard MD and replica exchange (REX) simulations.

**Figure supplement 4.** Accumulative deposition of history-dependent biases along the first three collective variables (CVs) for the replica-averaged metadynamics (RAM) simulations of the apo PKA-C.

3). The metadynamics simulations were equilibrated and converged in 300 ns (*Figure 2A*, *Figure 2—figure supplement 4*). The full free energy landscape was then reconstructed by sampling an extra 100 ns of production phase with reduced bias along each CV. We found that the population of the PKA-C conformers is modulated by ATP and pseudosubstrate binding within the NMR detection limit of sparsely populated states (~0.5% or $\Delta G < 3.2$ kcal/mol).

Specifically, the apo PKA-C populates preferentially a GS and five readily accessible low-populated excited states (*Figure 2B*, *Figure 2—source data 1*). The binary complex populates the same GS and a broad higher energy basin (*Figure 2C*, *Figure 2—source data 1*). Finally, the ternary complex

occupies a narrower dominant GS (*Figure 2D*, *Figure 2—source data 1*). Overall, these calculations revealed that the apo kinase explores multiple minima along each CV. The binding of ATP shifts the population of the ensemble into an additional minimum, featuring structures that are committed to substrate binding. Finally, the conformational heterogeneity of the ensemble is significantly reduced upon formation of the ternary complex, which represents a catalytically committed state (*Bastidas et al., 2013*; *Figure 2A*). The free energy landscape obtained from these RAM simulations is consistent with the qualitative picture previously inferred from our NMR spin relaxation experiments and hydrogen/deuterium exchange studies (*Masterson et al., 2011*; *Masterson et al., 2010*; *Li et al., 2015*), while providing a detailed structural characterization of the ground and sparsely populated conformationally excited states.

## MSM reveals the conformational transitions of PKA-C from ground to high free energy (excited) states

To refine the free energy landscape and delineate the kinetics of conformational transitions of the kinase, we performed additional unbiased sampling to build MSMs (*Kohlhoff et al., 2014*; *Plattner et al., 2017*). MSMs are commonly used to describe the dynamic transitions of various metastable states for macromolecules in terms of their populations and transition rates. MSMs are typically created by combining thousands of short unbiased simulations (*Kohlhoff et al., 2014*; *Plattner et al., 2017*). Following this strategy, we performed several short simulations (10–20 ns) using thousands of the low free energy conformations ($\Delta G < 3.2$ kcal/mol) chosen randomly from the three forms of PKA-C as starting structures. The conformational ensembles were clustered into microstates and seeded to start a second round of adaptive sampling (see Methods). This iterative process was repeated three times to assure convergence and yielded 100 µs trajectories for both the apo and binary forms, whereas 60 µs trajectories were collected for the less dynamic ternary complex. Once we reached a sufficient sampling, we built an MSM including L95, V104, L106, M118, M120, Y164, and F185 to investigate the dynamic transitions of the hydrophobic R spine and shell residues (*Figure 3—figure supplement 1*). These residues are ideal reporters of the dynamic processes governing the activation/deactivation of the kinase (*Kim et al., 2017*). To compare the free energy landscape of different complexes, we projected the conformational ensembles of three forms and existing crystal structures of PKA-C onto the first two time-lagged independent components (tICs) of the apo form, which were obtained by a time-lagged independent component analysis (tICA). These tICs represent the directions of the slowest motion of the kinase and visualize the conformational transitions of V104, L95, and F185 (*Figure 3—figure supplement 1*). The MSM shows that the hydrophobic core of the apo PKA-C accesses three major basins. The broadest basin represents the GS and corresponds to the conformations of PKA-C captured in several crystal structures (*Figure 3A*). The other two states (conformationally excited states) are less populated. The first excited state (ES1) features a disrupted hydrophobic packing of L95, V104, and F185. This conformation matches an inactive state with an outward orientation of the αC-helix typical of the inhibited states of PKA-C (*Figure 3—figure supplement 2*). The second state (ES2) displays a flipped configuration of the V104 side chain and disrupted hydrophobic interactions responsible for anchoring the αC-β4 loop to the C-lobe, with F100 and F102 adopting a gauche+ configuration and forming a stable π–π stacking. (*Figure 3A*). In contrast, the active GS ensemble features a *trans* configuration for the F100 and F102 side chains that stabilize the π–π stacking with Y156 at the αE-helix and cation-π interactions with R308 at the C-tail (*Figure 3A*). To our knowledge, the ES2 state was never observed in the available crystal structures. Upon ATP binding, the conformational space spanned by the kinase becomes narrower, and the conformers populate mostly the GS, with a small fraction in the ES1 state (*Figure 3B*). This is consistent with the role of the nucleotide as an allosteric effector that enhances the affinity of the enzyme for the substrate (*Masterson et al., 2008*). In the ternary form (ATP and PKI-bound), PKA-C populates only the GS consistent with the competent conformation observed in the ternary structure (i.e., 1ATP). In this case, the αC-β4 loop of the enzyme is locked in a well-defined, active configuration (*Figure 3A*).

Moreover, in the apo form, the αC-β4 loop is quite dynamic due to transient hydrophobic interactions involving F100 and V104, as well as W222 and the APE motif (A206 and P207) (*Figure 4A*). The binding of both nucleotide and PKI rigidifies the residues near F100 and V104, for example, V103 and F185 (*Figure 4A*) and makes more persistent several electrostatic interactions essential for catalysis (D166–N171, K168–T201, and Y204–E230), which are transient in the apo PKA-C. Finally, we used

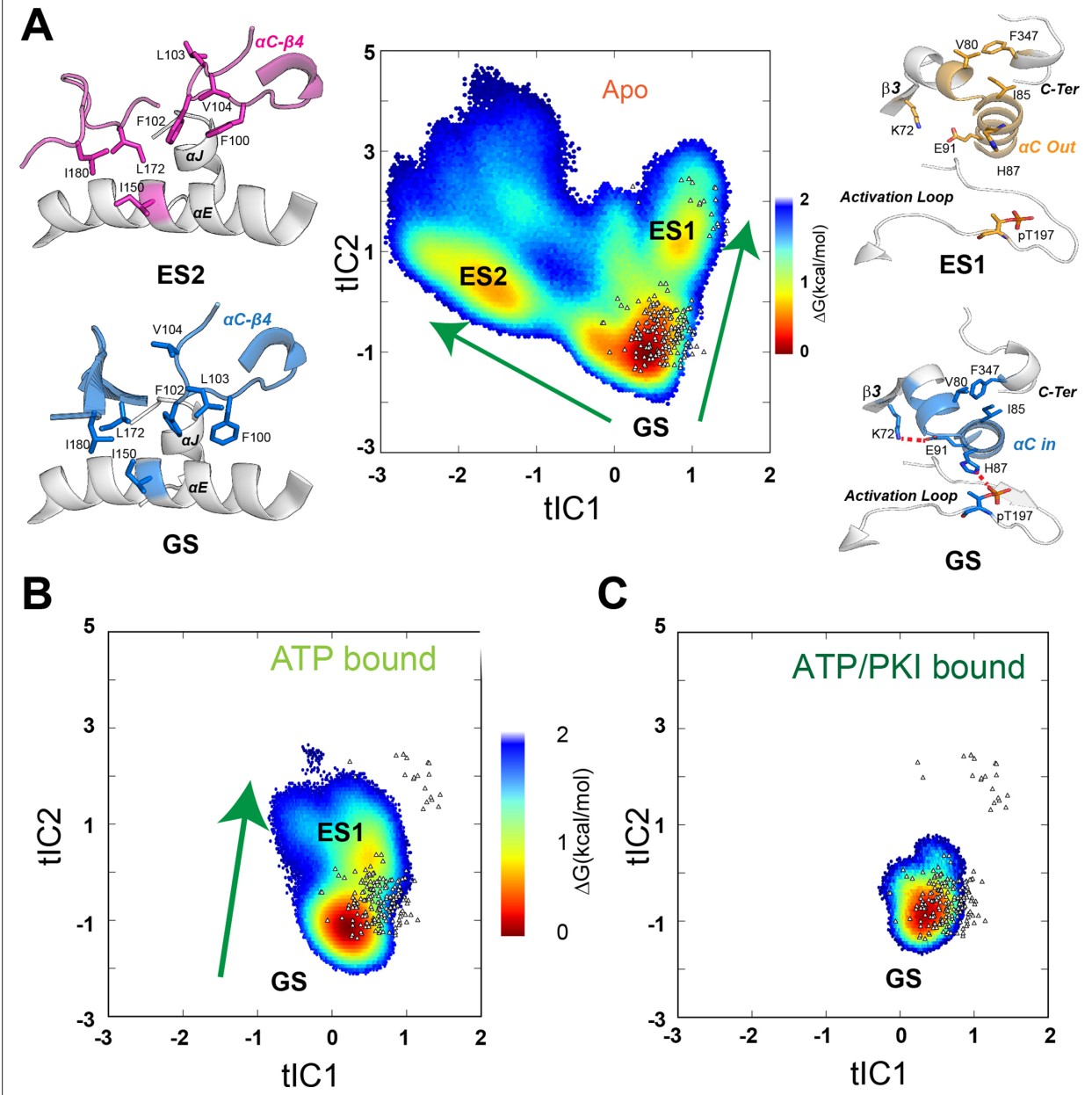

**Figure 3.** Free energy surfaces and dynamic transitions determined by a Markov State Model (MSM) for apo, ATP-, and ATP/PKI-bound PKA-C. (**A**) Free energy landscape projected along the first two time-lagged independent components (tICs) of apo PKA-C, featuring three basins, ground state (GS), ES1, and ES2. The transition from GS to ES1 (arrow) highlights the changes around the αB-αC loop, with the disruption of the K72–E91 salt bridge and the PIF pocket (V80–I85–F347) hydrophobic interactions. The GS to ES2 transition (arrow) displays the rearrangement of the hydrophobic packing around the αC-β4 loop. (**B, C**) Free energy surfaces projected along the first two tICs for the ATP- and ATP/PKI-bound PKA-C, respectively. Known crystal structures for the three forms are indicated by small white triangles.

The online version of this article includes the following figure supplement(s) for figure 3:

**Figure supplement 1.** R spine and shell residues selected for two time-lagged independent components (tICA) and Markov State Model (MSM) analysis.

**Figure supplement 2.** Structural features of αC-out transition (ES1) in various inactive kinases.

**Figure supplement 3.** Distinct hydrophobic packing for residues around the αC-β4 loop in the ground state (GS) and ES states of apo PKA-C.

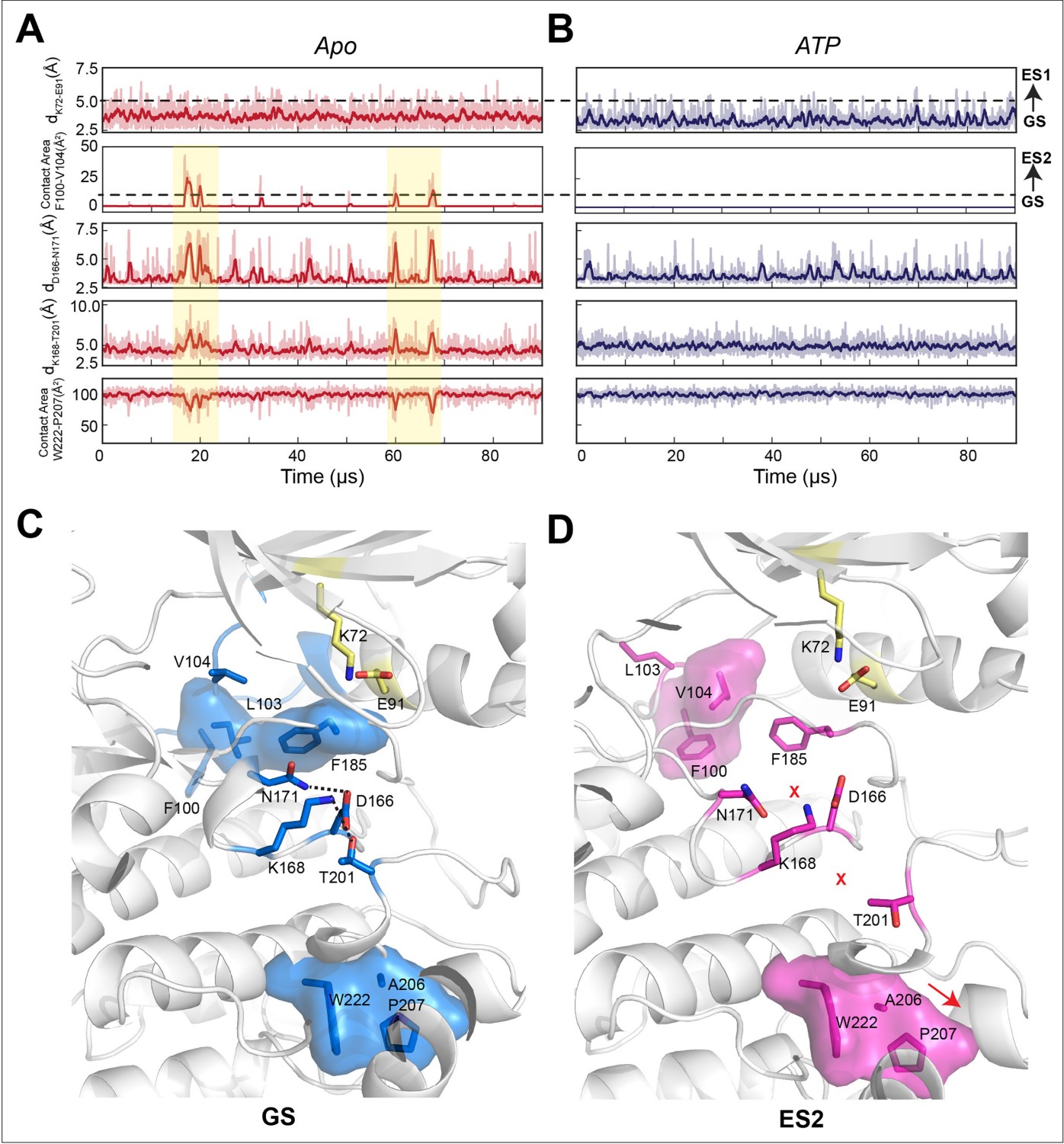

**Figure 4.** Conformational transitions of apo and ATP-bound PKA-C from ground state (GS) to ES1 and ES2 states as shown by the kinetic Monte Carlo trajectories. (**A, B**) Time course of the structural transitions from GS to ES1 and GS to ES2 for apo and ATP-bound PKA-C, respectively. The GS to ES1 transition is characterized by the disruption of the K72–E91 salt bridge and occurs frequently for both PKA-C forms. In contrast, the structural transition from GS to ES2 occurs only for the apo PKA-C, and it features the interactions between F100 and V104 that cause allosteric changes between D166–N171, K168–T201, and W222–A206–P207. The dark color traces indicate the moving averages calculated every 10 frames. (**C**) Structural snapshot of the GS conformation showing that the key catalytic motifs are poised for phosphoryl transfer. (**D**) Structural snapshot of the ES2 conformation with a disrupted configuration of key catalytic motifs typical of inactive kinase.

kinetic Monte Carlo sampling to characterize the slow transition between different states (*Shukla et al., 2014*). In the GS state of the apo PKA-C, F100 and F102 adopt *trans* configurations that stabilize the interactions with the αE and αJ helices, and, together with the nucleotide, they lock the αC-β4 loop in an active state (*Figure 3A*). The GS to ES1 transition features the disruption of the K72–E91 salt bridge, typical of the inactive kinase. In contrast, the GS to ES2 transition involves a 120° flip of the F100 aromatic group that interacts with V104, a conformation found only in the catalytically uncommitted apoenzyme (*Figure 3A*). Interestingly, the GS to ES2 transition involves a concerted disruption of the D166–N171 and K168–T201 electrostatic interactions and the destabilization of the packing between W222 and the APE motif (A206 and P207) required for substrate recognition (*Figure 3C*). Overall, these calculations show that conformational transitions from the active GS to ES1 and ES2 represent two independent pathways toward inactive states of the kinase.

## Direct correspondence between the conformationally excited states identified by MD simulations and NMR data

NMR CPMG relaxation dispersion and CEST experiments performed on the apo PKA-C revealed the presence of conformationally excited states for several residues embedded into the hydrophobic core of the enzyme (*Olivieri et al., 2022*). The CSs of one of these states follow the previously observed open-to-closed transitions of the enzyme (*Masterson et al., 2008*), whereas another excited state observable in the CEST experiments does not follow the same trend. We surmised that this new state may represent an alternate inactive state (*Olivieri et al., 2022*). Since ES2 features a disrupted hydrophobic packing of the αC-β4 loop and several of the methyl groups displaying the excited state in the CEST experiments are near the αC-β4 loop, we compared the CS differences of the methyl group from CEST experiments (*Olivieri et al., 2022*) ($\Delta\omega^{Exp}$) with those from the MD simulations ($\Delta\omega^{Pred}$). We analyzed the methyl groups of L103, V104, I150, L172, and I180, sampling more than 500 snapshots as representative conformations of the kinase in the ES2 and GS states (*Figure 3—figure supplement 3*). We then computed the distribution of the methyl $^{13}C$ CS using ShiftX2 (*Han et al., 2011*). For Val104 and Ile150, these calculations yielded $\Delta\omega^{Pred}$ values of 0.87 ± 0.02 and 0.85 ± 0.02 ppm between the ES2 and GS conformations, respectively, which are in good agreement with $\Delta\omega^{Exp}$ (1.10 ± 0.06 and 0.94 ± 0.09 ppm) obtained from the CEST experiments or fitting the CPMG dispersion curves (*Figure 5A, B*). The $\Delta\omega^{Pred}$ values and the directions of the CS changes obtained for the remainder three sites are also in good agreement with the CEST profiles (*Figure 5—figure supplement 1*). *Figure 4C* shows a linear fitting of $\Delta\omega^{Pred}$ and $\Delta\omega^{Exp}$ with a slope of 0.86 and $R^2$ of 0.82. The latter supports the hypothesis that the excited state observed by NMR may correspond to the simulated structural ensemble of the ES2 basin. Finally, MSM estimates a population of ES2 of 6 ± 2%, which is consistent with the 5 ± 1% population found by fitting the NMR data (*Kim et al., 2017*). Altogether, the combination of NMR and MD simulations support the presence of an alternate inactive state that features disrupted hydrophobic interactions near the αC-β4 loop that perturb the anchoring with the αE helix, and in turn, the structural couplings between the two lobes of the kinase.

## The F100A mutation disrupts the allosteric network of the kinase

The above analysis suggests that the destabilization of the αC-β4 loop may promote the GS to ES2 transition of the kinase. To validate this hypothesis, we generated the F100A mutant seeking to increase the flexibility of the αC-β4 loop. Starting from the coordinates of the X-ray structure of the ternary complex (PDB ID: 4WB5), we simulated the F100A dynamics in an explicit water environment. First, we performed a short equilibration using classical MD simulations. During 1 μs of MD simulations, the αC-β4 loop of the F100A mutant undergoes a significant motion as manifested by increased values of the backbone root mean square deviation (rmsd) and a conformational change (flipping) of the F102 side chain (*Figure 6A*). This region in PKA-C^WT adopts a stable β-turn, with a persistent hydrogen bond between the backbone oxygen of F100 and the amide hydrogen of L103. In contrast, the hydrogen bond in F100A is formed more frequently between A100 and F102, resulting in an average γ-turn conformation (*Figure 6B*). Such local rearrangement not only disrupts the hydrogen bond between N99 and Y156, altering the interactions between the αC-β4 loop and the αE helix but also destabilizes the cation-π interaction between F102 and R308 at the C-tail (*Figure 6B—figure supplement 1*).

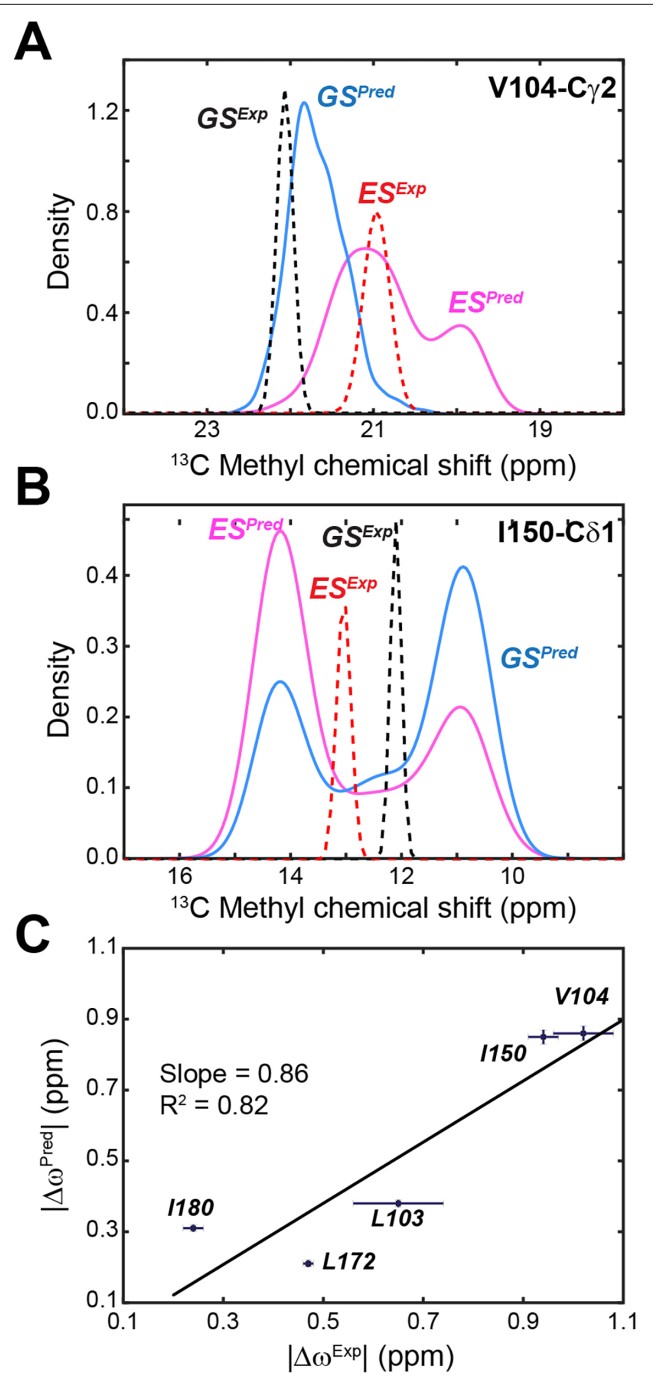

**Figure 5.** Comparison of experimental versus calculated $^{13}C$ chemical shifts (CSs) of methyl groups of PKA-C. (**A**) Experimental and calculated CS for Val104-Cγ1 of apo PKA-C. The ground state (GS) is in blue and the ES in magenta. (**B**) Corresponding CS profiles for Ile150-Cδ1. The experimental CS is shown in dotted lines for GS (black) and ES (red). (**C**) Correlation between predicted $|\Delta\omega^{Pred}|$ and experimental $|\Delta\omega^{Exp}|$ CS differences for methyl groups near the αC-β4 loop. The fitted linear correlation has a slope of 0.86 and $R^2$ of 0.82. The experimental errors were estimated from the signal-to-noise ratios of the CEST spectra.

The online version of this article includes the following figure supplement(s) for figure 5:

**Figure supplement 1.** Distribution of predicted and experimental $^{13}C$ chemical shift (CS) of selected methyl groups.

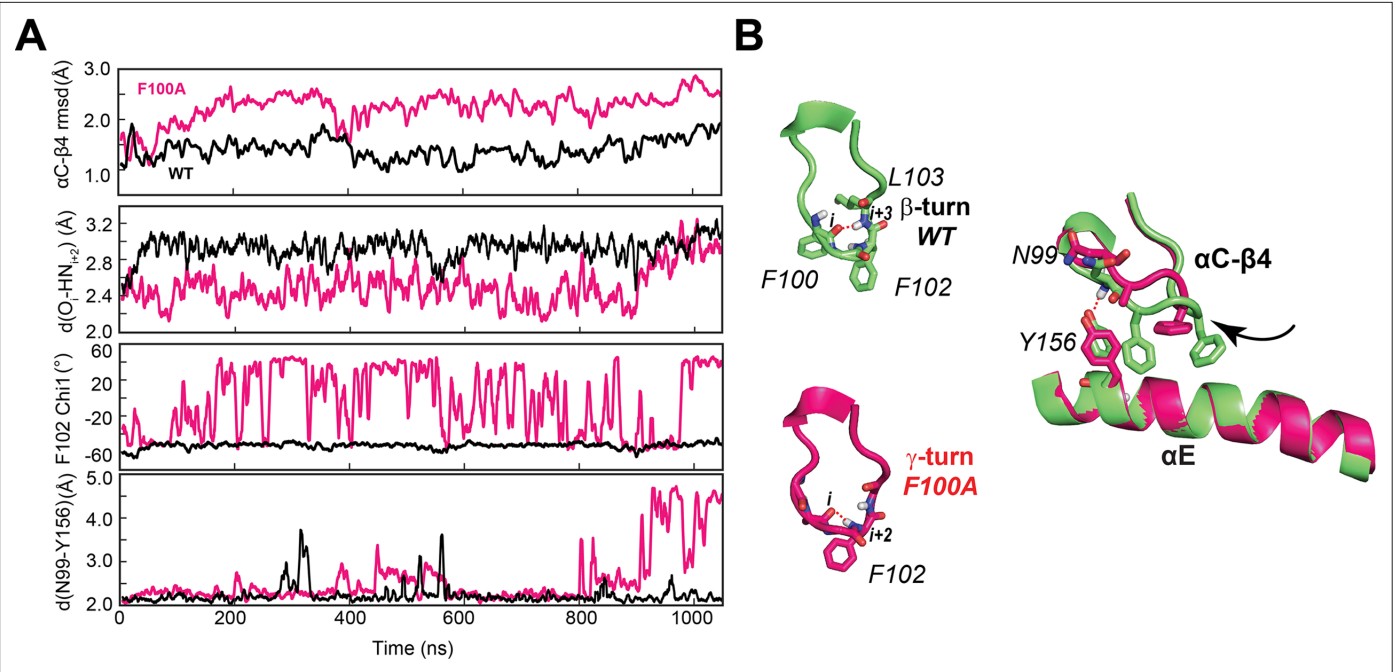

**Figure 6.** F100A mutation increases the dynamics of the αC-β4 loop, perturbing the local hydrophobic packing and its anchoring to the αE helix. (**A**) Time series of the αC-β4 loop dynamics, H-bond occurrence for the β- and γ-turns, F102 $\chi_1$ angle, and N99 and Y156 for WT (black) and F100A (red) in the ATP-bound state. (**B**) Representative structural snapshots showing the β-turn conformation for PKA-C^WT (green) and γ-turn for PKA-C^F100A (magenta).

The online version of this article includes the following figure supplement(s) for figure 6:

**Figure supplement 1.** Time series of the distance between F102 and R308 for PKA-C^WT (black) and the PKA-C^F100A mutant (magenta).

The structural changes of the αC-β4 loop caused by the F100A mutation propagate through the protein core across the R spine, C spines, and shell residues (**Figure 7A**), and alter the response of the kinase to nucleotide binding. In the PKA-C^WT, ATP-binding shifts the conformational ensemble of the kinase toward an intermediate state competent to substrate binding, as represented by the changes in the population densities of the hydrophobic core residues as a function of the rmsd (**Figure 7B**). For F100A, the population of the C spine follows the same trend of the WT kinase, with the shell residues already populating the intermediate state. In contrast, the R spine residues fail to adopt a competent state (**Figure 7B**). The latter is due to the perturbation of hydrophobic packing of L95 and L106 in the R spine, and V104 in the shell near the mutation site. Using the lowest principal components, we also analyzed the global dynamic response of the kinase to ATP binding. Not only does F100A change the breathing mode of the two lobes (PC1), but it also alters the shearing motion (PC2) of the binary complex, emphasizing the importance of this allosteric site for the inter-lobe communication (**Figure 7C–E**).

To analyze the internal correlated motions of the kinase, we further analyzed the MD trajectories of both wild-type and mutant enzyme using mutual information (**Figure 8**; **McClendon et al., 2012**). This method identifies correlated changes of backbone and side chain rotamers in proteins and detects clusters of residues that are responsible for intramolecular communication between active sites (**McClendon et al., 2012**). For PKA-C^WT, we observed numerous strong correlations that are clustered within each lobe and across the enzyme, connecting key motifs, such as the Gly-rich loop, αC-β4 loop, activation loop, and C-terminal tail (**Figure 8A**). These correlation patterns are reminiscent of the CS correlations experimentally found for PKA-C^WT (**Wang et al., 2019**), which constitute the hallmark of the dynamically committed state (**Masterson et al., 2011**; **Masterson et al., 2010**). In contrast, the mutual information matrix for the F100A mutant displays weaker and interspersed correlations across the enzyme. New correlations also are present across the helical C-lobe suggesting a rewiring of the internal allosteric communication (**Figure 8B**). A possible explanation is that the increased motion caused by the elimination of the F100 aromatic side chain at the αC-β4 loop propagates throughout

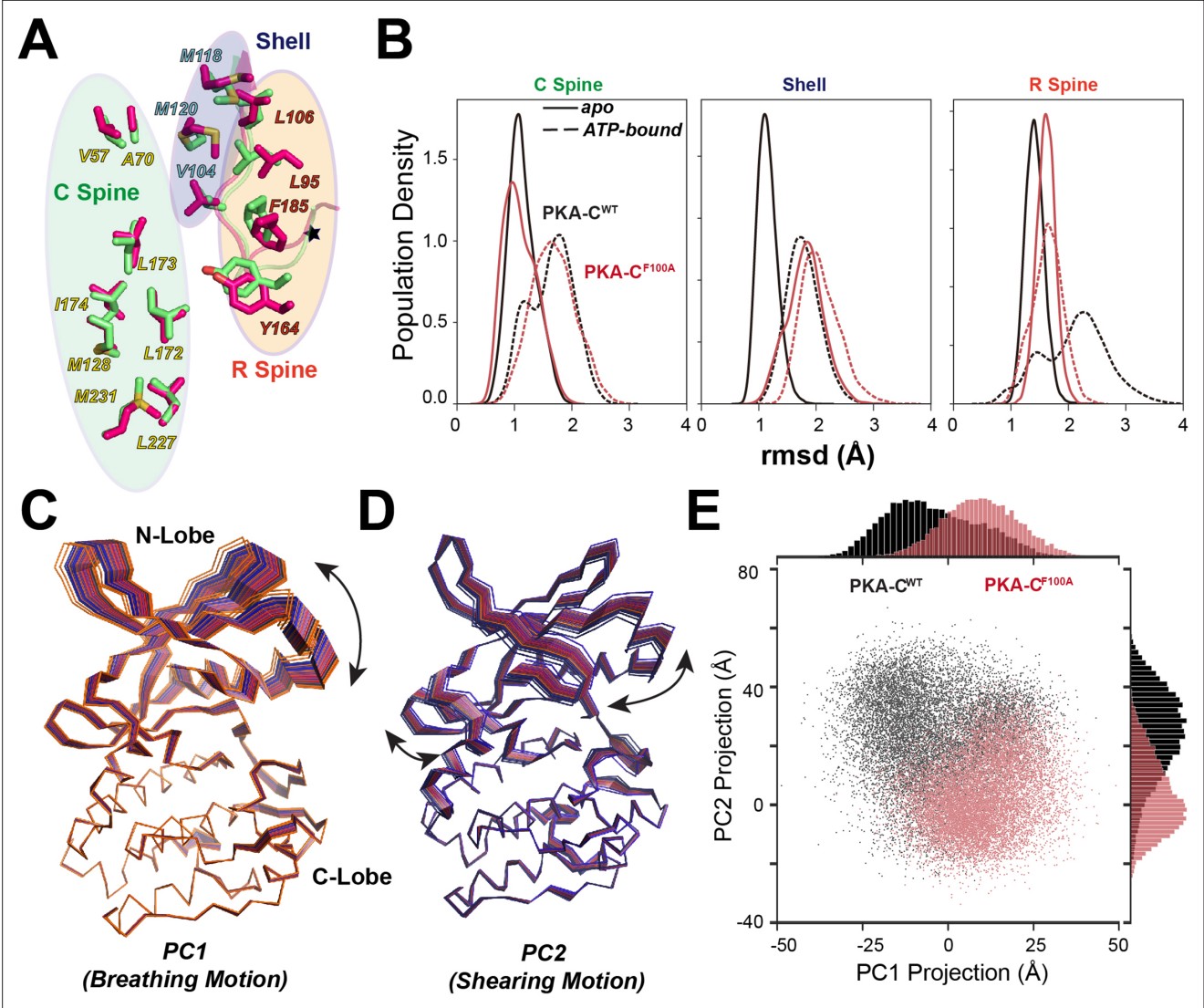

**Figure 7.** Structural responses to ATP binding of PKA-C$^{F100A}$ mutant. (**A**) Superposition of the hydrophobic cores (C spine, R spine, and shell residues) for PKA-C$^{WT}$ (lime) and PKA-C$^{F100A}$ (hot pink), highlighting the structural perturbations of the R spine and shell residues. (**B**) Structural perturbation upon ATP binding for the hydrophobic core residues of PKA-C$^{WT}$ and PKA-C$^{F100A}$ shown as changes in the population densities versus rmsd. (**C, D**) First (PC1) and second (PC2) principal components describing breathing and shearing motions of the two lobes. (**E**) 2D projections and distributions of PC1 and PC2 for PKA-C$^{WT}$ and PKA-C$^{F100A}$.

the entire hydrophobic core. This structural reorganization causes the F100A kinase to adopt a dynamically uncommitted state (*Masterson et al., 2011*). Taken together, the simulations of F100A suggest that perturbations of the αC-β4 loop increase the local flexibility and break the structural connection between the two lobes, disrupting the correlated breathing/shearing motions highlighted in previous MD simulation studies (*Tsigelny et al., 1999*; *Lu et al., 2005*). Although the single F100A mutation does not drive the kinase into a completely inactive state (ES2), it is sufficient to abolish the structural couplings between the two lobes.

Based on the MD simulations of the F100A mutant, we hypothesized that the disruption of the structural coupling between the N- and C-lobe would affect the binding cooperativity between nucleotide (ATPγN) and pseudosubstrate inhibitor (PKI$_{5-24}$). To test this hypothesis, we expressed, purified, and evaluated the catalytic efficiency for the WT and F100A kinases by carrying out steady-state coupled enzyme assays (*Cook et al., 1982*) using the standard substrate Kemptide (*Kemp et al., 1977*). F100A showed only a slight increase in $K_M$ and $V_{max}$ compared to PKA-C$^{WT}$, resulting in a small reduction of the catalytic efficiency ($k_{cat}/K_M$ = 0.50 ± 0.04 and 0.41 ± 0.08 for PKA-C$^{WT}$ and PKA-C$^{F100A}$,

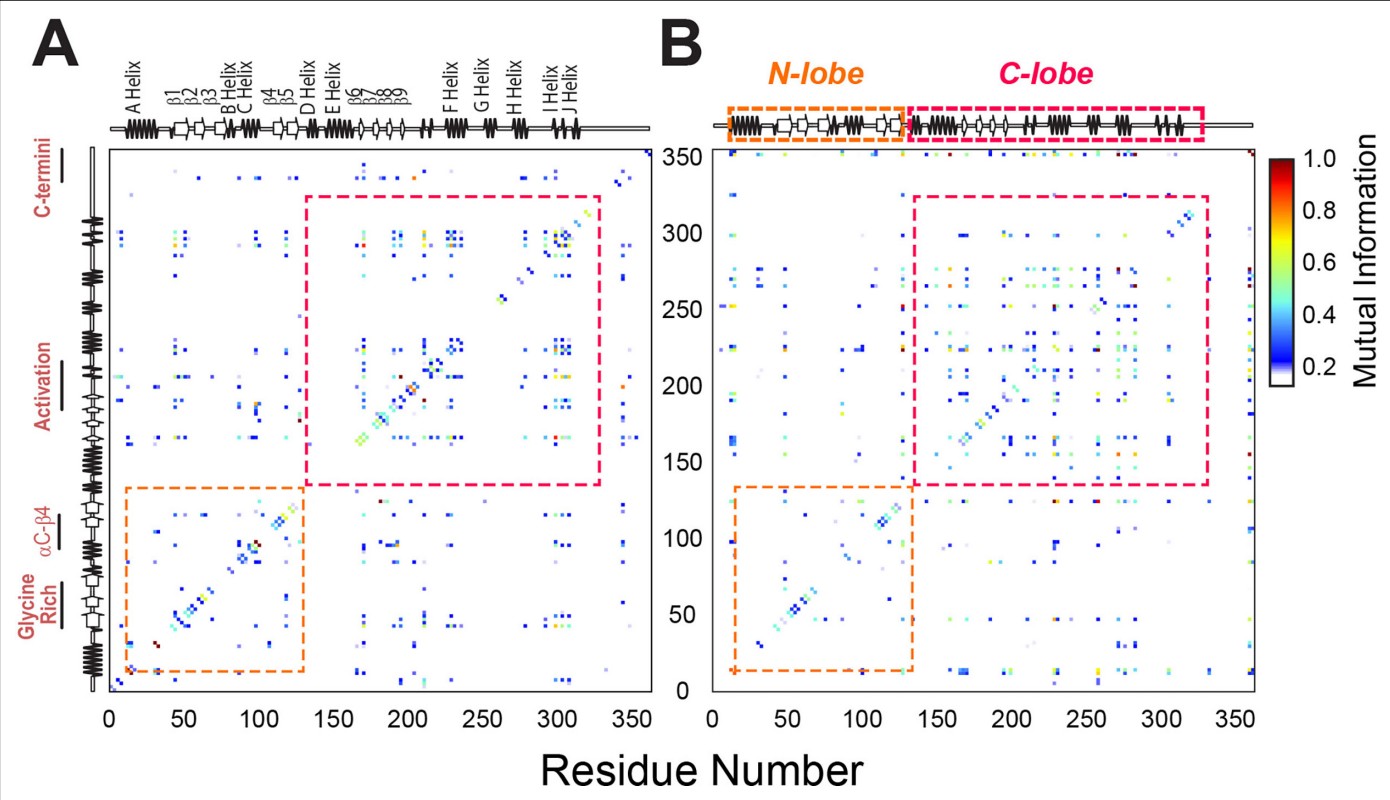

**Figure 8.** Mutual information analysis of backbone and side chain rotamers of ATP-bound PKA-C$^{WT}$ and PKA-C$^{F100A}$. (**A**) Mutual information matrix for PKA-C$^{WT}$ showing well-organized clusters of interactions within each lobe and distinct inter-lobe communication typical of an active kinase. (**B**) Mutual information matrix for PKA-C$^{F100A}$ revealing an overall reorganization of the allosteric network caused by the disruption of the hydrophobic core. Effects of the F100A mutation on the catalytic efficiency and binding thermodynamics of PKA-C.

respectively; *Supplementary file 1*). We then performed ITC (*Wiseman et al., 1989*) to obtain ΔG, ΔH, −TΔS, and $K_d$, and determine the cooperativity coefficient (σ) for ATPγN and PKI$_{5-24}$ binding. We first analyzed the binding of ATPγN to the apo PKA-C$^{F100A}$ and, subsequently, the binding of PKI$_{5-24}$ to the ATPγN-saturated PKA-C$^{F100A}$ (*Supplementary file 2*). We found that PKA-C$^{WT}$ and PKAC$^{F100A}$ have similar binding affinities for ATPγN ($K_d$ = 83 ± 8 and 73 ± 2 μM, respectively). However, in the apo form, F100A showed a threefold higher binding affinity for the pseudosubstrate relative to PKA-C$^{WT}$ ($K_d$ = 5 ± 1 and 17 ± 2 μM, respectively – *Supplementary file 2*). Upon saturation with ATPγN, PKA-C$^{F100A}$ displayed a 12-fold reduction in binding affinity for PKI$_{5-24}$, resulting in a σ of ~3, a value significantly lower than the WT enzyme (σ greater than 100). These data support the predictions of MD simulations and mutual information that ATP binding to F100A does not promote a conformational state fully competent with substrate binding.

## NMR mapping of nucleotide-/pseudosubstrate-binding response

To elucidate the atomic details of the disrupted structural coupling and binding cooperativity for PKA-C$^{F100A}$, we used solution NMR spectroscopy mapping its response to the nucleotide (ATPγN) and pseudosubstrate (PKI$_{5-24}$) binding. Specifically, we monitored the $^1$H and $^{15}$N chemical shift perturbations (CSPs, Δδ) of the amide fingerprints for the binary (PKA-C$^{F100A}$/ATPγN) and ternary (PKA-C$^{F100A}$/ATPγN/PKI$_{5-24}$) complexes and compared them with the corresponding complexes of PKA-C$^{WT}$ (*Figure 9* and *Figure 9—figure supplement 1*). The interaction with ATPγN causes a dramatic broadening of several amide resonances throughout the structure of PKA-C$^{F100A}$, suggesting the presence of an intermediate conformational exchange as previously observed (*Srivastava et al., 2014*). For the remainder of the amide peaks, both PKA-C$^{WT}$ and PKA-C$^{F100A}$ exhibit similar CSP profiles upon binding ATPγN (*Figure 9A, B*), although the extent of the CS changes is significantly attenuated for F100A (*Figure 9—figure supplement 1B*). This reduction is apparent for residues in the N-lobe (β2–β3

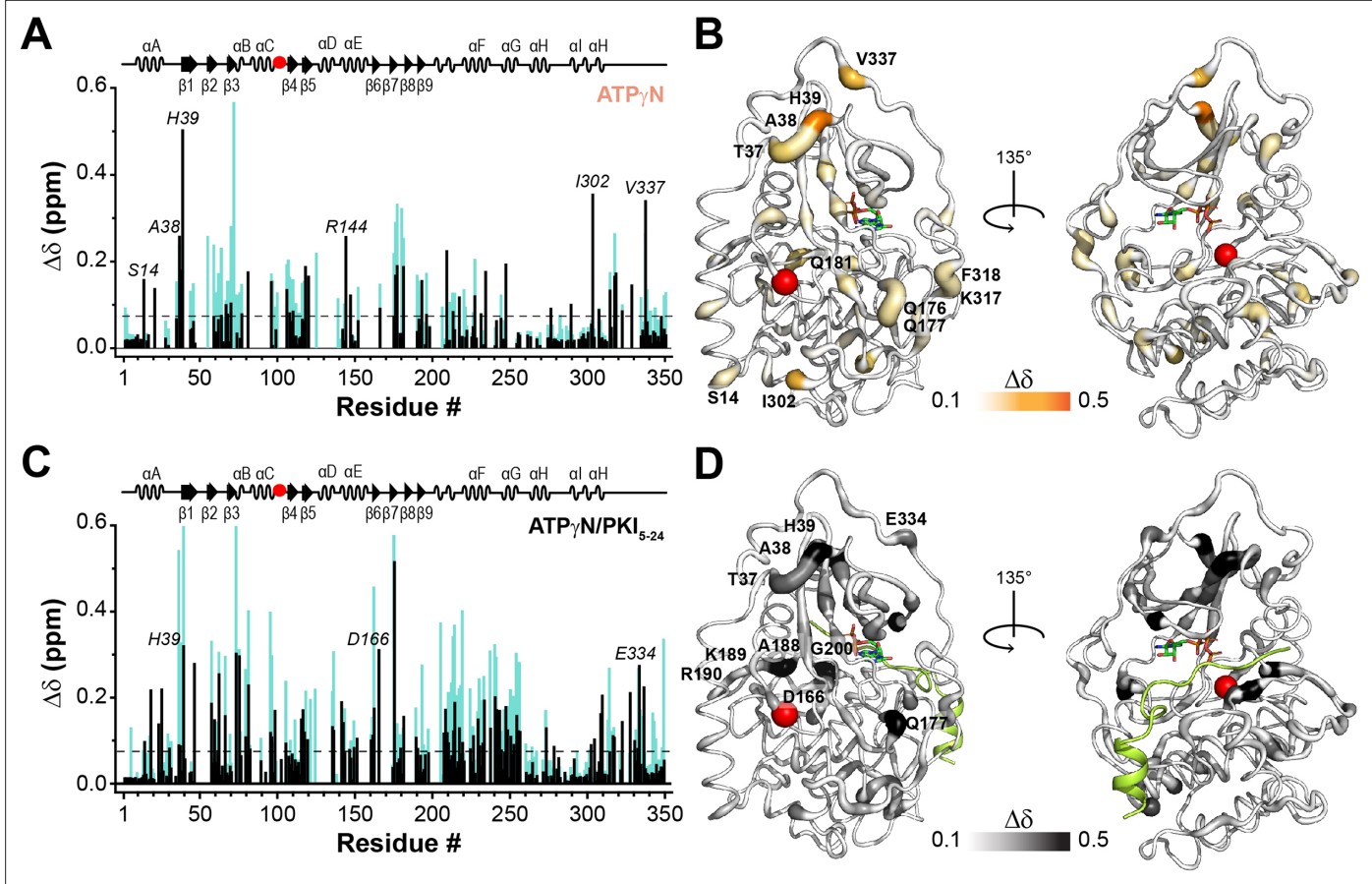

**Figure 9.** NMR map of the structural response of PKA-C^F100A to nucleotide and protein kinase inhibitor (PKI) binding. (**A**) Comparison of the chemical shift perturbation (CSP) of the amide resonances for PKA-C^F100A (black) and PKA-C^WT (cyan) upon ATPγN binding. The dashed line indicates one standard deviation from the average CSP. (**B**) CSPs of PKA-C^F100A/ATPγN amide resonances mapped onto the crystal structure (PDB: 4WB5). (**C**) Comparison of the CSPs of the amide resonances for PKA-C^F100A and PKA-C^WT upon binding ATPγN and PKI_{5-24} (black). (**D**) CSPs for the F100A/ATPγN/PKI complex mapped onto the crystal structure (PDB: 4WB5).

The online version of this article includes the following figure supplement(s) for figure 9:

**Figure supplement 1.** NMR fingerprints of PKA-C^F100A.

**Figure supplement 2.** COordiNated ChemIcal Shifts bEhavior (CONCISE) plot showing the shifts of the probability distribution of the amide resonances as a function of nucleotides and substrate binding.

region), around the mutation site (β4), and at the C-terminal tail. A similar pattern emerges upon binding PKI_{5-24} to the ATPγN-saturated PKA-C^F100A, where a substantial decrease in CSP is observed for residues of the C-lobe localized near the motifs critical for substrate binding (i.e., αE, αF, and αG helices, *Figure 9C, D*).

To determine the global response to ligand binding for WT and F100A, we analyzed the CSs of the amide fingerprints of the two proteins using the COordiNated ChemIcal Shifts bEhavior (CONCISE) (*Cembran et al., 2014*). CONCISE describes the overall changes of the protein fingerprint resonances by providing the probability density (population) of each state along the conformational equilibrium for binding phenomena that follow linear CS trajectories (*Cembran et al., 2014*). For PKA-C^WT, both nucleotide- and pseudosubstrate-binding shift the overall population of the amides from an open to an intermediate and a fully closed state (*Figure 9—figure supplement 2*). A similar trend is observed for PKA-C^F100A, though the probability densities are broader, indicating that the amide resonances follow a less coordinated response (*Cembran et al., 2014*). Also, the maximum of the probability density for the ternary complex is shifted toward the left, indicating that the mutant adopts a more open structure than the corresponding wild-type enzyme. Overall, the shapes and the positions of the probability distributions of the resonances suggest that several residues do not respond in

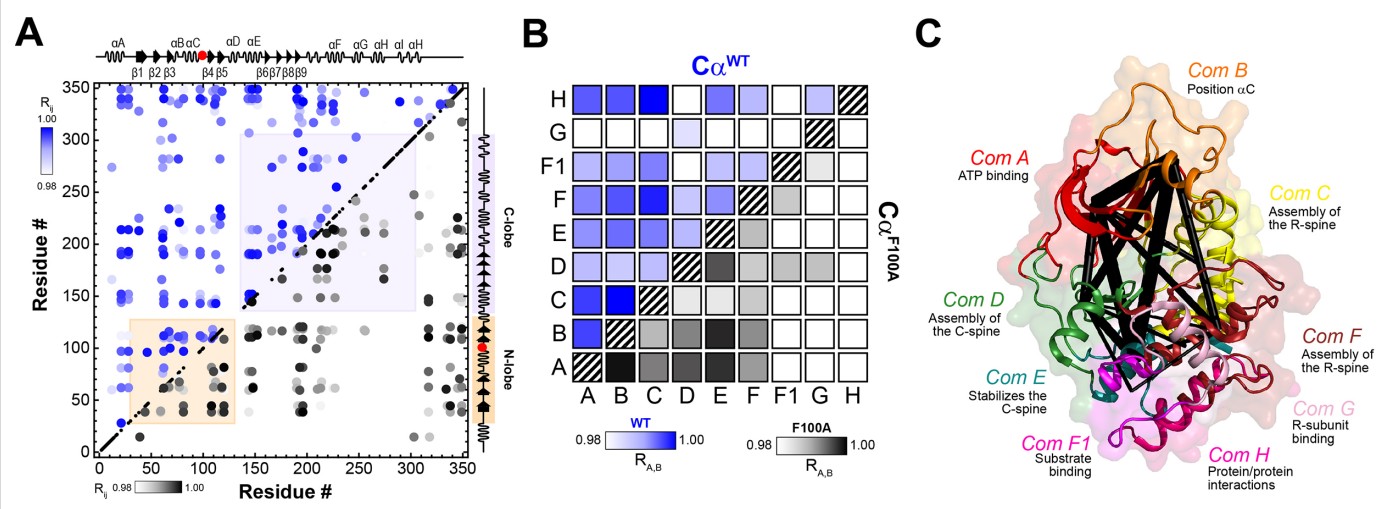

**Figure 10.** Correlated chemical shift changes reveal the uncoupling of the intramolecular allosteric network in PKA-C$^{F100A}$. (**A**) Comparison of the CHEmical Shift Covariance Analysis (CHESCA) matrices obtained from the analysis of the amide CS of PKA-C$^{WT}$ (blue correlations) and PKA-C$_{F100A}$ (black correlations). The correlations coefficients ($R_{ij}$) were calculated using the apo, ADP-, ATPγN-, and ATPγN/PKI$_{5-24}$-bound states. For clarity, only correlation with $R_{ij} > 0.98$ are displayed. For the enlarged CHESCA map of F100A see *Figure 10—figure supplement 1*. The data for the PKA-C$^{WT}$ matrix were taken from *Walker et al., 2019*. (**B**) Community CHESCA analysis of PKA-C$^{WT}$ (blue correlations) and PKA-C$^{F100A}$ (black correlations). Only correlations with $R_{A,B} > 0.98$ are shown. (**C**) Spider plot showing the extent of intramolecular correlations identified by the community CHESCA analysis for PKA-C$^{F100A}$ mapped onto the crystal structure (PDB: 4WB5). The thickness of each line in the spider plot indicates the extent of coupling between the communities.

The online version of this article includes the following figure supplement(s) for figure 10:

**Figure supplement 1.** Intermolecular allosteric network of F100A mapped using CHEmical Shift Covariance Analysis (CHESCA) and community CHESCA.

(**A**) CHESCA matrix obtained from the amide chemical shift trajectories of PKA-C$^{F100A}$ in the apo, ADP-, ATPγN-, and ATPγN/PKI$_{5-24}$-bound states. Only correlations with $R_{ij} > 0.98$ are displayed. (**B**) Plot of the correlation scores versus residue calculated for PKA-C$^{WT}$ (blue) and PKA-C$^{F100A}$ (black). (**C**) Community CHESCA analysis of PKA-C$^{F100A}$. Only correlations with $R_{A,B} > 0.98$ are shown. (**D**) Spider plots indicating the correlated structural communities of PKA-C$^{F100A}$ and PKA-C$^{WT}$ plotted on their corresponding structures. The size of each node is independent of the number of residues it encompasses, and the weight of each line indicates the strength of coupling between the individual communities.

a coordinated manner to the nucleotide binding, and PKI$_{5-24}$ shifts the conformation of the kinase toward a partially closed state, which explains the loss in binding cooperativity as previously observed (*Olivieri et al., 2021*; *Walker et al., 2019*; *Walker et al., 2021*).

To define the allosteric network of the kinase upon binding nucleotides and substrate, we examined the CS using CHEmical Shift Covariance Analysis (CHESCA) (*Walker et al., 2019*; *Walker et al., 2021*; *Olivieri et al., 2022*), a statistical method that identifies correlated responses of residue pairs to a specific perturbation (i.e., ligand binding, mutations, etc.). CHESCA works under the assumption that pairwise correlated CS changes of residues identify possible intramolecular allosteric networks (*Akimoto et al., 2013*; *Selvaratnam et al., 2011*). For PKA-C, we found that coordinated structural rearrangements, as identified by CHESCA, are directly related to the extent of binding cooperativity (*Walker et al., 2019*; *Walker et al., 2021*; *Olivieri et al., 2022*). Therefore, we compared the CHESCA maps for PKA-C$^{WT}$ and PKAC$^{F100A}$ for four different states: *apo*, ATPγN-, ADP-, and ATPγN/PKI$_{5-24}$-bound. For PKA-C$^{F100A}$, the CHESCA matrix exhibits sparser and more attenuated correlations (i.e., lower correlation coefficient value) relative to PKA-C$^{WT}$ (*Figure 10A*). Although many inter-lobe correlations are still present for F100A, several other correlations in specific structural domains such as the αG-, αH-, and αI-helices are absent or attenuated. For instance, the F100A mutation does not display correlations between the αA-helix and the C-terminal tail that constitute a critical 'complement to the kinase core' (*Veron et al., 1993*). We also utilized CHESCA to assess the allosteric communication among the PKA-C communities as defined by *McClendon et al., 2014*. The CHESCA community map for PKA-C$^{WT}$ shows strong correlations across the enzyme, especially for structurally adjacent communities and at the interface between the two lobes (see for instance the correlations among *ComA, ComB, ComC, ComE, and ComH*) (*Figure 10B, C*). For F100A, the CHESCA community map shows that the cross-talk between the nucleotide-binding (*ComA*) and positioning of

αC-helix (*ComB*) communities, as well as the R-spine assembly (*ComC*) and the activation loop (*ComF*) communities are preserved (*Figure 10B, C*). However, the correlations between *ComE*, responsible for stabilizing the C spine, and *ComC*, involved in the assembly of the R spine, are absent. Similarly, the long-range correlations between the C and R spines (i.e., *ComD* with *ComC*) are missing. Finally, several correlations between *ComF1*, *ComG*, and *ComH* are no longer present. These communities orchestrate substrate recognition and R subunits binding. Overall, the CHESCA analysis for PKA-C$^{F100A}$ suggests that the reduced degree of cooperativity we determined thermodynamically corresponds to a decrease in coordinated structural changes upon ligand binding. The latter is apparent from the loss of correlated structural changes among the structural communities, including the hydrophobic spines, substrate binding cleft, and the docking surface for PKA interactions with other binding partners.

## Discussion

Our structural and dynamic studies suggest that the binding cooperativity of PKA-C originates from the allosteric coupling between the nucleotide-binding pocket and the interfacial region between the two lobes, which harbors the substrate binding cleft (*Olivieri et al., 2021*; *Walker et al., 2019*; *Walker et al., 2021*). The latter has been emphasized for other kinases such as Src and ERK2 (*Foda et al., 2015*; *Iverson et al., 2020*). The biological significance of this intramolecular communication has emerged from our recent studies on pathological mutations situated in the activation loop of PKA-C and linked to Cushing's syndrome (*Olivieri et al., 2021*; *Walker et al., 2019*; *Walker et al., 2021*). These mutations drastically reduce substrate-binding affinity, and more importantly, disrupt the communication between the two ligand binding pockets of the kinase (*Olivieri et al., 2021*; *Walker et al., 2019*; *Walker et al., 2021*). Interestingly, the E31V mutation distal from the active site and also related to the Cushing's syndrome, displays disfunction similar to the other orthosteric mutations, suggesting that it is possible to modulate substrate recognition allosterically (*Walker et al., 2021*). Indeed, these mutations are connected to allosteric nodes that, once perturbed, radiate their effects toward the periphery of the enzyme interrupting the coordinated dynamic coupling between the two lobes (*Walker et al., 2021*). Interestingly, these mutations do not prevent Kemptide phosphorylation; rather, they cause a loss of substrate fidelity with consequent aberrant phosphorylation of downstream substrates (*Omar et al., 2022*; *Beuschlein et al., 2014*; *Calebiro et al., 2017*; *Calebiro et al., 2014*; *Di Dalmazi et al., 2014*; *Lubner, 2017*). Additionally, recent thermodynamic and NMR analyses of PKA-C binding to different nucleotides and inhibitors demonstrated that it is possible to control substrate-binding affinity by changing the chemistry of the ligand at the ATP-binding pocket (*Walker et al., 2019*). Altogether, these studies put forward the idea that substrate phosphorylation and cooperativity between ATP and substrates may be controlled independently.

By integrating our NMR data with RAM simulations and MSM, we comprehensively mapped the free energy landscape of PKA-C in various forms. We found that the active kinase unleashed from the regulatory subunits occupies a broad energy basin (GS) that corresponds to the conformation of the ternary structure of PKA-C with ATP and pseudosubstrate (PKI$_{5-24}$), exemplifying a catalytic competent state poised for phosphoryl transfer (*Gerlits et al., 2019*). We also identified two orthogonal conformationally excited states, ES1 and ES2. While ES1 corresponds to canonical inactive kinase conformations, the ES2 state was never observed in the crystallized structures. Our previous CEST NMR measurements suggested the presence of an additional sparsely populated state that, at that time, we were unable to characterize structurally (*Olivieri et al., 2022*). The CSs of this state did not follow the opening and closing of the kinase active cleft and were assigned to a possible alternate inactivation pathway (*Olivieri et al., 2022*). These new simulations and MSM show that this sparsely populated state may be attributed to the transition from GS to ES2, which features a disruption of hydrophobic packing, with a conformational rearrangement for the αC-β4 loop that causes a partial disruption of the hydrophobic R spine. These structural changes rewire the allosteric coupling between the two lobes, as shown by mutual information analysis. A single mutation (F100A), suggested by our simulations, promotes the flip of the αC-β4 loop and reproduces the hypothesized structural uncoupling between the two lobes of PKA-C. We experimentally tested the effects of the F100A mutation and found that it prevents the enzyme from adopting a conformation competent for substrate binding, resulting in a drastic reduction of the cooperativity between ATP and substrate. These NMR data further support our working model, showing that if the inter-lobe communication is interrupted, the binding response of the kinase is attenuated. Indeed, our investigation further emphasizes the integral role of the kinase

hydrophobic core and demonstrates that alterations in the spines and shell residues may lead to a dysfunctional kinase by either preventing phosphorylation (see V104G and I150A mutations) (*Kim et al., 2017*) or by disrupting the binding cooperativity as for F100A.

The αC-β4 loop is a regulatory element present in all EPKs and its importance has been stressed in bioinformatics studies (*Kannan and Neuwald, 2005*) and supported by computational work (*McClendon et al., 2014*). F100 and F102 in the αC-β4 loop are at intersections of various structural communities and constitute a critical hydrophobic node that anchors the N- to the C-lobe. Also, studies on EGFR(epidermal growth factor receptor) and ErbB2 kinases led to the hypothesis that the αC-β4 loop may act as a molecular brake (*Chen et al., 2007*; *Klein et al., 2015*) or an autoinhibitory switch (*Fan et al., 2008*). Therefore, it is not surprising that activating mutations and in-frame insertions in the αC-β4 loop are frequently found in kinase-related cancers and somatic oncogenic mutations such as P101S, P101L, and L103F (*McSkimming et al., 2015*). Also, an elegant study by Kannan and coworkers emphasized the role of the αC-β4 loop in dimerization and aberrant activation of EGFR by insertion mutations (*Ruan and Kannan, 2018*). Moreover, in a recent paper Zhang and co-workers found a similar behavior for the HER2 exon 20 insertions (*Zhao et al., 2020*). These latter two studies propose that in-frame insertions alter the conformational landscape of the kinase rigidifying the structure around the αC-β4 loop and restricting the kinase conformational ensemble in a constitutively active state. Remarkably, in the case of EGRF insertion mutations, the activation of the kinase occurs in a length-dependent manner (*Ruan and Kannan, 2018*). Additionally, these human mutations display a gradual response to drugs, which can be exploited for designing selective inhibitors against EGFR pathological mutants (*Yeung et al., 2020*).

The data presented here show that it is possible to abolish the binding cooperativity of a kinase by turning the dial in the opposite direction, that is, increasing the flexibility of the αC-β4 loop and disconnecting the allosteric network between the N- and C-lobes. The identification of this new, partially inactivating pathway provides further understanding of how to control the dynamics and function of kinases.

# Materials and methods

**Key resources table**

| Reagent type (species) or resource | Designation | Source or reference | Identifiers | Additional information |
|---|---|---|---|---|
| Gene (*Homo sapiens*) | PKA-Cα | | PKA-C | Uniprot ID: P17612 |
| Strain, strain background (*Escherichia coli*) | BL21(DE3) pLyss | Agilent | Cat. #200132 | Chemically competent cells |
| Sequence-based reagent | PKA-C$^{F100A}$ | This study | PCR primer (Forward) | tattctgcaagcggtgaacg cccc gtttctggttaagctg |
| Sequence-based reagent | PK I α 5-24 | Synthetic peptide | PKI$_{5-24}$ | Chemically synthesized |
| Sequence-based reagent | Kemptide | Synthetic peptide | Kemptide | LRRASLG |
| Commercial assay or kit | QuikChange Lightning Multi Mutagenesis Kit | Agilent genomics | Cat #210519 | Commercial mutagenesis kit |
| Recombinant DNA reagent | PKA-$^{CF100A}$ | This study | PKA-C$^{F100A}$ | Single Ala mutant of PKA-C |
| Chemical compound, drug | AMP-PNP or ATPγN | Roche Applied Science | CAS 25612-73-1 | ATP analogous |
| Chemical compound, drug | ADP | Sigma-Aldrich | CAS 20398-34-9 | Nucleotide |
| Software, algorithm | TopSpin 4.1 | Bruker Inc | https://www.bruker.com/ | |
| Software, algorithm | NMRFAM-Sparky | NMRFAM | https://nmrfam.wisc.edu/nmrfam-sparky-distribution/ | |
| Software, algorithm | NMRPipe | Delaglio F., NIH | https://www.ibbr.umd.edu/nmrpipe/install.html | |

*Continued on next page*

| Reagent type (species) or resource | Designation | Source or reference | Identifiers | Additional information |
|---|---|---|---|---|
| Software, algorithm | POKY | Lee W. | https://sites.google.com/view/pokynmr | |
| Software, algorithm | COordiNated ChemIcal Shift bEhavior (CONCISE) | Veglia G. | https://conservancy.umn.edu/handle/11299/217206 https://conservancy.umn.edu/handle/11299/227294 | Matlab script |
| Software, algorithm | CHEmical Shift Covariance Analysis (CHESCA) | Melacini G. | https://academic.oup.com/bioinformatics/article/37/8/1176/5905475?login=true | NMRFAM-Sparky &POKY tools |
| Software, algorithm | PyMol | Schrödinger, LLC | https://pymol.org | |
| Software, algorithm | MatLab2022b | MathWorks | https://www.mathworks.com/products/matlab.html | |
| Software, algorithm | GraphPad Prism 9 | GraphPad Software Inc | https://www.graphpad.com/ | |
| Software, algorithm | GROMACS 4.6 | Hess B et al. | http://www.gromacs.org/ | |
| Software, algorithm | CHARMM36a1 | *Best et al., 2012* | https://doi.org/10.1021/ct300400x | |
| Software, algorithm | PLUMED 2.1.1 | *Bonomi et al., 2009* | https://www.plumed.org/doc-v2.5/user-doc/html/_c_h_a_n_g_e_s-2-1.html | |
| Software, algorithm | ALMOST 2.1 | *Kohlhoff et al., 2009*. | https://svn://svn.code.sf.net/p/almost/code/almost-code | |
| Software, algorithm | METAGUI | *Biarnés et al., 2012* | https://www.sciencedirect.com/science/article/pii/S0010465511003079 | |
| Software, algorithm | MDTraj | *McGibbon et al., 2015* | https://www.sciencedirect.com/science/article/pii/S0006349515008267 | |
| Software, algorithm | SPARTA+ | *Shen and Bax, 2010* | https://spin.niddk.bov/bax/software/SPARTA+/ | |
| Software, algorithm | MSMbuilder | *Harrigan et al., 2017*; | https://msmbuilder.org/ | |
| Software, algorithm | Mutinf | *McClendon et al., 2012* | https://simtk.org/projects/mutinf/ | |

## RAM simulations

### System setup

As a starting structure, we used the coordinates of the crystal structure of PKA-C$^{WT}$ (PDB ID: 1ATP) and added the missing residues 1–14 at the N terminus. The protonation state of histidine residues was set as previously reported (*Cembran et al., 2012*). The kinase was solvated in a rhombic dodecahedron solvent box using the three-point charge TIP3P model (*Jorgensen et al., 1983*) for water, which extended approximately 10 Å away from the surface of the proteins. Counter ions (K$^+$ and Cl$^-$) were added to ensure electrostatic neutrality to a final ionic concentration of ~150 mM. All protein covalent bonds were constrained using the LINCS algorithm (*Hess et al., 1997*) and long-range electrostatic interactions were simulated using the particle-mesh Ewald method with a real-space cutoff of 10 Å (*Darden et al., 1993*). The simulations of apo, binary (one Mg$^{2+}$ ion and one ATP), and ternary (two

Mg²⁺ ions, one ATP, and PKI₅₋₂₄) forms were performed simultaneously using GROMACS 4.6 (*Hess et al., 1997*) with CHARMM36a1 force field (*Best et al., 2012*). For F100A, the corresponding residues were mutated through the mutagenesis wizard of Pymol (*DeLano, 2002*).

## Standard MD simulations

Each system was minimized using the steepest descent algorithm to remove the geometric distortions, and then gradually heated to 300 K at a constant volume over 1 ns using harmonic restraints with a force constant of 1000 kJ/(mol*Å) (*Huse and Kuriyan, 2002*) on heavy atoms for both the kinase and nucleotide. The restraints were gradually released over the following 12 ns of simulations at constant pressure (1 atm) and temperature (300 K). The systems were equilibrated for an additional 20 ns without positional restraints. A Parrinello–Rahman barostat (*Parrinello and Rahman, 1980*) was used to keep the pressure constant, while a V-rescale thermostat (*Bussi et al., 2007*) with a time step of 2 fs was used to keep the temperature constant. Each system was simulated for 1.05 μs, with snapshots recorded every 20 ps.

## Replica exchange (REX) simulations

Following standard MD simulations, parallel REX simulations were set up on the apo, binary, and ternary forms of PKA-C. Four replicas were used for each REX simulation, and the initial structures were randomly chosen from the μs-scale unbiased simulations. CSs of PKA-C for N, CA, CO, CB, and HN from NMR experiments were imposed as restraints based on the following penalty function:

$$E^{cs} = \alpha \sum_{k=1}^{N} \sum_{l=1}^{5} \left( \delta_{kl}^{exp} - \frac{1}{M} \sum_{m=1}^{M} \delta_{kl}^{calc} \right)^2$$

where $\alpha$ is the force constant, $k$ runs over all residues of the protein, $l$ denotes the different backbone atoms, and $m$ runs over the four replicas. $\delta_{kl}^{calc}$ was computed using CamShift, a module of ALMOST-2.1 (*Kohlhoff et al., 2009*). The force constant was gradually increased from 0 (unbiased) to 20 (maximum restraints for production) over 50 ns. All other settings were identical to the unbiased simulations. REX simulations were carried out with GROMACS 4.6 (*Hess et al., 2008*), with the REX controlled by the PLUMED 2.1.1 module (*Bonomi et al., 2009*). Approximately, 100 ns of REX simulations were further carried out for each replica in the three PKA-C forms.

## RAM simulations

RAM simulations were started from the final structures of REX simulations. The CS restraints were imposed in the same manner as for the REX simulations. Four CVs are chosen to increase the conformational plasticity around the catalytic cores (detailed in *Figure 2—figure supplement 1*): (CVI) the $\psi$ angles of the backbone of all the loops that are not in contact with ATP (Back-far), (CVII) the ϕ angles of the backbone of all the loops that are in contact with ATP (Back-close), (CVIII) the $\chi_1$ angles of side chains of all the loops that are in contact with ATP (Side-close), (CVIV) the radius of gyration calculated over the rigid part (i.e., residues 50–300) of the protein (rgss). Gaussian deposition rate was performed with an initial rate of 0.125 kJ/mol/ps, where the σ values were set to 0.5, 0.2, 0.2, and 0.01 nm for the four CVs, respectively. The RAM simulations were also carried out with GROMACS 4.6 in conjunction with PLUMED 2.1 and ALMOST 2.1, and continued for ~400 ns for each replica with exchange trails every 1 ps.

## Reconstruction of free energy surface (FES) from the RAM simulations

After about 300 ns, the sampling along the first three CVs reached convergence with fluctuations of bias within 1 kcal/mol, allowing a reliable reconstruction of the corresponding FES. The production run was continued for another 100 ns to sample enough conformations. These conformations were first clustered into microstates using the regular spatial method (cutoff radius of 0.13), and then the free energy of each state was reweighted according to the deposited potential along each CV, which can be obtained from the analysis module of METAGUI (*Biarnés et al., 2012*). To visualize the distribution of these microstates and their relative energy differences, we further performed principal component analysis to project the microstates represented in the three-dimensional CV space into

two-dimensional space spanned by PC1 and PC2. Then we plotted these microstates in a three-dimensional space spanned by PC1, PC2, and free energy differences $\Delta G$.

## Independent validation of CSs with SPARTA+

During the REX and RAM simulations, the CSs were computed via CamShift in ALMOST 2.1 (*Kohlhoff et al., 2009*). As an independent validation for the efficacy of the bias, we further calibrated the CSs of 2000 snapshots with MDTraj (*McGibbon et al., 2015*) and SPARTA+ (*Shen and Bax, 2010*).

## Adaptive sampling

The first round of adaptive sampling started from the snapshots of low-energy microstates obtained from the previous step, that is, 1200 structures for the apo form, 400 structures for the binary form, and 200 structures for the ternary form. The initial velocities were randomly generated to satisfy the Maxwell distribution at 300 K. For the apo form, a 10-ns simulation was performed for each run, whereas for the binary, each simulation lasted 30 ns, resulting in a total of 12 µs trajectories for both the apo and binary forms. Three rounds of adaptive sampling were started from the 400 microstates, derived via *K*-mean clustering of all snapshots of previous ensembles, to obtain a converged free energy landscape. Therefore, a total of 100 µs trajectories and 1,000,000 snapshots (100 ps per frame) were collected for both the apo and binary form after three rounds of adaptive sampling, and a total of 60 µs trajectories were collected for the ternary form.

## MSM and tICA

The Cartesian coordinates of key hydrophobic residues, including R-spine residues, L95, L106, Y164, and F185, and the shell residues, V104, M118, and M120, were chosen as the metrics to characterize the conformational transition of the hydrophobic core of PKA-C. Specifically, each snapshot was first aligned to the same reference structure by superimposing the αE (residues 140–160) and αF helices (residues 217–233) and represented by the deviation of the Cartesian coordinates of key residues. The representation in this metric space was further reduced to 10-dimension vectors using tICA (*Schwantes and Pande, 2013*) at a lag time of 1 ns. All snapshots were clustered into 400 microstates with *K*-mean clustering. An MSM was built upon the transition counts between these microstates.

## Kinetic Monte Carlo trajectory of PKA-C in different forms

Long trajectories were generated using a kinetic Monte Carlo method based on the MSM transition probability matrix of the three forms of PKA-C. Specifically, the discrete jumps between the 100 microstates were sampled for 60 µs. Random conformations were chosen for each state from all the available snapshots. Subsequently, the time series of various order parameters were analyzed and plotted.

## Protein expression and purification

The recombinant human Cα subunit of cAMP-dependent protein kinase with the Phe to Ala mutation in position 100 (PKA-C[F100A]) was generated from the human PKA-Cα wild-type using QuikChange Lightning mutagenesis kit (Agilent genomics). The key resource table lists the PCR primers used to modify the pET-28a expression vector encoding for the wild-type human PKA-Cα gene (*PRKACA* – uniprot P17612) (*Olivieri et al., 2021*; *Walker et al., 2019*; *Walker et al., 2021*). The unlabeled and uniformly [15]N-labeled PKA-C[F100A] mutant was expressed and purified following the same protocols used for the wild-type protein (*Masterson et al., 2008*). Briefly, transformed *E. coli* BL21 (DE3) *pLyss* cells (Agilent) were cultured overnight at 30°C in Luria-Bertani (LB) medium. The next morning, the cells were transferred to fresh LB medium for the overexpression of the unlabeled protein or to M9 minimal medium supplied with [15]NH$_4$Cl (Cambridge Isotope Laboratories Inc) as the only nitrogen source for the labeled protein overexpression. In both cases, protein overexpression was induced with 0.4 mM of β-D-thiogalactopyranoside and carried out for 16 hr at 20°C. The cells were harvested by centrifugation and resuspended in 50 mM Tris–HCl, 30 mM KH$_2$PO$_4$, 100 mM NaCl, 5 mM 2-mercaptoethanol, 0.15 mg/ml lysozyme, 200 µM ATP, DNaseI, 1 tablet of cOmplete ULTRA EDTA-free protease inhibitors (Roche Applied Science) (pH 8.0) and lysed using French press at 1000 psi. The cell lysate was cleared by centrifugation (60,000 × *g*, 4°C, 45 min), and the supernatant was batch-bound with Ni$^{2+}$-NTA agarose affinity resin (Thermo Scientific). The his-tagged PKA-C[F100A]

was eluted with 50 mM Tris–HCl, 30 mM $KH_2PO_4$, 100 mM NaCl, 5 mM 2-mercaptoethanol, 0.5 mM phenylmethylsulfonyl fluoride (PMSF) (pH 8.0) supplied with 200 mM of imidazole. The tail of poly-His was cleaved using a stoichiometric amount of recombinant tobacco etch virus protease in 20 mM $KH_2PO_4$, 25 mM KCl, 5 mM 2-mercaptoethanol, 0.1 mM PMSF (pH 6.5), overnight at 4°C. The different phosphorylation states of PKA-C$^{F100A}$ were separated using a cation exchange column (HiTrap Q-SP, GE Healthcare Life Sciences) using a linear gradient of KCl in 20 mM $KH_2PO_4$ at pH 6.5 (*Yonemoto et al., 1993*). The purified protein isoforms were then stored in phosphate buffer containing 10 mM dithiothreitol (DTT), 10 mM $MgCl_2$, and 1 mM $NaN_3$ at 4°C. The protein purity was assessed by sodium dodecyl sulfate–polyacrylamide gel electrophoresis.

## Peptide synthesis

The Kemptide (LRRASLG) and PKI$_{5-24}$ (TTYADFIASGRTGRRNAIHD) peptides were synthesized using a CEM Liberty Blue microwave synthesizer using standard Fmoc chemistry. All peptides were cleaved with Reagent K (82.5% trifluoroacetic acid, 5% phenol, 5% thioanisole, 2.5% ethanedithiol, and 5% water) for 3 hr and purified using a semipreparative Supelco C18 reverse-phase HPLC column at 3 ml/min. The purified peptides were concentrated, lyophilized, and stored at −20°C. Molecular weight and quantity were verified by matrix assisted laser desorption ionization-time of flight mass spectrometry (MALDI-TOF) and/or amino-acid analysis (Texas A&M University).

## ITC measurements

PKA-C$^{F100A}$ was dialyzed into 20 mM 3-(*N*-morpholino)propanesulfonic acid (MOPS), 90 mM KCl, 10 mM DTT, 10 mM $MgCl_2$, and 1 mM $NaN_3$ (pH 6.5) and concentrated using conical spin concentrator (10 kDa membrane cutoff, Millipore) to a solution at 80–100 µM, as confirmed by A280 = 55,475 M$^{-1}$ cm$^{-1}$. Approximately 300 µl of protein was used for each experiment, with 50 µl of 2 mM ATPγN and/or 1 mM PKI$_{5-24}$ in the titrant syringe. All measurements were performed at 300 K in triplicates with a low-volume NanoITC (TA Instruments). The binding was assumed to be 1:1, and curves were analyzed with the NanoAnalyze software (TA Instruments) using the Wiseman isotherm (*Wiseman et al., 1989*).

$$\frac{d\left[MX\right]}{d\left[X_{tot}\right]} = \Delta H^\circ V_0 \left[\frac{1}{2} + \frac{1 - \frac{1 - r}{2} - R_m/2}{\left(R_m^2 - 2R_m\left(1 - r\right) + \left(1 + r\right)^2\right)^{1/2}}\right]$$

where $d[MX]$ is the change in total complex relative to the change in total protein concentration, $d[X_{tot}]$ is dependent on $r$ (the ratio of $K_d$ relative to the total protein concentration), and $R_m$ (the ratio between total ligand and total protein concentration). The heat of dilution of the ligand into the buffer was considered for all experiments and subtracted.

The free energy of binding was determined from:

$$\Delta G = RTlnK_d$$

where $R$ is the universal gas constant and $T$ is the temperature at measurement (300 K). The entropic contribution to binding was calculated using:

$$T\Delta S = \Delta H - \Delta G$$

The degree of cooperativity ($\sigma$) was calculated as:

$$\sigma = \frac{K_{dapo}}{K_{dnucleotide}}$$

where $K_{d\,apo}$ is the dissociation constant of PKI$_{5-24}$ binding to the apo-enzyme, and $K_{d\,nucleotide}$ is the corresponding dissociation constant for PKI$_{5-24}$ binding to the nucleotide-bound kinase.

## Enzyme assays

Steady-state coupled enzyme activity assays using Kemptide as substrate were performed under saturating ATP concentrations and spectrophotometrically at 298 K, as reported by *Cook et al., 1982*. The values of $V_{max}$ and $K_M$ were obtained from a nonlinear fit of the initial velocities to the Michaelis–Menten equation.

## NMR spectroscopy

NMR measurements were performed on a Bruker Avance NEO spectrometer operating at a $^1$H Larmor frequency of 600 MHz equipped with a cryogenic probe or on a Bruker Avance III 850 MHz spectrometer equipped with a TCI cryoprobe. The NMR experiments were recorded at 300 K in 20 mM KH$_2$PO$_4$ (pH 6.5), 90 mM KCl, 10 mM MgCl$_2$, 10 mM DTT, 1 mM NaN$_3$, 5% D$_2$O, and 0.1% 4-benzene sulfonyl fluoride hydrochloride (AEBSF, Pefabloc – Sigma-Aldrich). Concentrations for samples were 0.15 mM of uniformly $^{15}$N-labeled PKA-C$^{F100A}$, as determined by $A_{280}$ measurements, 12 mM ATPγN or ADP was added for the nucleotide-bound form, and 0.3 mM PKI$_{5-24}$ for the ternary complex. [$^1$H, $^{15}$N]-WADE-TROSY-HSQC pulse sequence (*Manu et al., 2022*) was used to record the amide fingerprint spectra of PKA-C$^{F100A}$ in the apo, nucleotide-bound (ADP- or ATPγN-bound – binary form), and ternary complex (PKAC$^{F100A}$/ATPγN/PKI$_{5-24}$). All [$^1$H, $^{15}$N]-WADE-TROSY-HSQC experiments were acquired with 2048 (proton) and128 (nitrogen) complex points, processed using NMRPipe (*Delaglio et al., 1995*) and visualized using NMRFAM-SPARKY (*Lee et al., 2015*) and POKY (*Manthey et al., 2022*). Combined CSPs were calculated using $^1$H and $^{15}$N CSs according to:

$$\Delta\delta = \sqrt{(\Delta\delta H)^2 + (0.154 \times \Delta\delta N)^2}$$

in which $\Delta\delta$ is the CSP; $\Delta\delta_H$ and $\Delta\delta_N$ are the differences of $^1$H and $^{15}$N CSs, respectively, between the first and last points of the titration; and 0.154 is the scaling factor for nitrogen (*Williamson, 2013*).

## COordiNated ChemIcal Shift bEhavior

The normal distributions reported in the CONCISE plot were calculated using principal component analysis for residues whose CSs responded linearly to ligand binding (*Cembran et al., 2014*). In this work, we use the $^1$H and $^{15}$N CSs derived from the [$^1$H, $^{15}$N]-WADE-TROSY-HSQC experiments for the apo, ADP-, ATPγN-, and ATPγN/PKI$_{5-24}$-bound forms of PKA-C.

## CHEmical Shift Covariance Analysis

This analysis was used to identify and characterize allosteric networks of residues showing concerted responses to nucleotide and pseudosubstrate binding. To identify inter-residue correlations, four states were used: apo, ATPγN-, ADP-, and ATPγN/PKI$_{5-24}$-bound. The identification of inter-residue correlations by CHESCA relies on agglomerative clustering and singular value decomposition (*Selvaratnam et al., 2011*). Pairwise correlations of CS changes were calculated to identify networks. When plotted on a correlation matrix, the 2D correlations allow the identification of local and long-range correlated residues. For each residue, the maximum change in the CS was calculated in both the $^1$H (*x*) and $^{15}$N (*y*) dimensions ($\Delta\delta_{x,y}$). The residues included in the CHESCA analysis were those satisfying $\Delta\delta_{x,y} > ½ \Delta v_{xA,yA} + ½ \Delta v_{xB,yB}$, where *A* and *B* correspond to two different forms analyzed, and $\Delta v$ denotes the linewidth. Correlation scores were used to quantify the CHESCA correlation of a single residue or a group of residues with another group. Correlation scores were evaluated for single residues or for the entire protein using the following expression:

$$Corr\,Score = \frac{number\,of\,(R_{ij} > R_{cutoff})}{total\,number\,of\,R_{ij}}$$

where $R_{ij}$ is the correlation coefficient and *i* and *j* denote (1) a single residue and the remainder residues of the protein, respectively, or (2) both represent all the assigned residues in the entire protein. For all the analyses a $R_{cutoff}$ of 0.98 was used. Community CHESCA analysis utilizes a similar approach to map correlations between functional communities within the kinase. Each community represents a group of residues associated with a function or regulatory mechanism (*McClendon et al., 2014*). To represent community-based CHESCA analysis, we utilized $R_{cutoff} > 0.8$. For instance, to represent a CS

correlation between two communities ($X$ and $Y$) with $n_A$ and $n_B$ number of assigned residues, respectively, the correlation score between $A$ and $B$ is defined as

$$R_{A,B} = number\ of\ (R_{ij} > R_{cutoff})\ /\ (n_A * n_B)$$

where $R_{ij}$ is the correlation coefficient between residue $i$ (belonging to community $A$) and residue $j$ (belonging to community $B$). $R_{cutoff}$ is the correlation value cutoff. $R_{A,B}$ assumes values from 0 (no correlation between residues in $A$ and $B$) to 1 (all residues in $A$ have correlations > cutoff with all residues in $B$).

## Materials availability statement

All newly created materials are available upon request. Details on how to obtain these materials can be found by contacting the corresponding author or through the repository website mentioned above. This ensures transparent disclosure and accessibility for further research and verification.

## Acknowledgements

This work was supported by the National Institutes of Health GM 100310 and HL 144130 to GV. The authors would like to acknowledge the Minnesota Supercomputing Institute for MD calculations. YW would like to thank the Guangdong Pearl River Talent Program (2021QN02Y618) and the National Natural Science Foundation of China (22007069, 92269102) for part of the MD analysis carried out at the Shenzhen Bay Laboratory Supercomputing Centre.

## Additional information

### Funding

| Funder | Grant reference number | Author |
| --- | --- | --- |
| National Institute of General Medical Sciences | GM100310 | Gianluigi Veglia |
| National Heart, Lung, and Blood Institute | NHLBI | Gianluigi Veglia |
| Guangdong Pearl River Talent Program | 2021QN02Y618 | Yingjie Wang |
| National Natural Science Foundation of China-China Academy of General Technology Joint Fund for Basic Research | 22007069 | Yingjie Wang |
| National Natural Science Foundation of China-China Academy of General Technology Joint Fund for Basic Research | 92269102 | Yingjie Wang |
| National Institutes of Health | 100310 | Gianluigi Veglia |
| National Institutes of Health | 144130 | Gianluigi Veglia |

The funders had no role in study design, data collection, and interpretation, or the decision to submit the work for publication.

### Author contributions

Cristina Olivieri, Conceptualization, Formal analysis, Validation, Investigation, Visualization, Writing – original draft, Writing – review and editing; Yingjie Wang, Conceptualization, Data curation, Software, Formal analysis, Validation, Investigation, Visualization, Writing – original draft; Caitlin Walker, Investigation, Writing – review and editing; Manu Veliparambil Subrahmanian, Data curation, Formal analysis,

Validation, Investigation, Visualization; Kim N Ha, Formal analysis, Writing – review and editing; David Bernlohr, Carlo Camilloni, Formal analysis, Supervision, Writing – review and editing; Jiali Gao, Formal analysis, Supervision, Funding acquisition, Writing – review and editing; Michele Vendruscolo, Formal analysis, Supervision, Writing – original draft, Writing – review and editing; Susan S Taylor, Conceptualization, Supervision, Funding acquisition, Writing – original draft, Writing – review and editing; Gianluigi Veglia, Conceptualization, Data curation, Supervision, Funding acquisition, Visualization, Writing – original draft, Project administration, Writing – review and editing

### Author ORCIDs
Cristina Olivieri (ID) https://orcid.org/0000-0001-6957-6743
Yingjie Wang (ID) http://orcid.org/0000-0001-9800-8163
Manu Veliparambil Subrahmanian (ID) http://orcid.org/0000-0002-1374-2797
Jiali Gao (ID) http://orcid.org/0000-0003-0106-7154
Carlo Camilloni (ID) https://orcid.org/0000-0002-9923-8590
Michele Vendruscolo (ID) http://orcid.org/0000-0002-3616-1610
Susan S Taylor (ID) http://orcid.org/0000-0002-7702-6108
Gianluigi Veglia (ID) http://orcid.org/0000-0002-2795-6964

Reviewer #1 (Public Review): https://doi.org/10.7554/eLife.91506.3.sa1
Reviewer #3 (Public Review): https://doi.org/10.7554/eLife.91506.3.sa2
Author response https://doi.org/10.7554/eLife.91506.3.sa3

## Additional files

### Supplementary files
• Supplementary file 1. Kinetic parameters of Kemptide phosphorylation by PKA-C$^{WT}$ and PKA-C$^{F100A}$ obtained from coupled assays. The $K_M$ and $V_{max}$ values were obtained from a nonlinear least squares analysis of the concentration-dependent initial phosphorylation rates. Errors in the $k_{cat}/K_M$ ratios were propagated from the individual errors in $K_M$ and $k_{cat}$.

• Supplementary file 2. Changes in enthalpy, entropy, free energy, and dissociation constants for nucleotide binding to PKA-C$^{WT}$ and PKA-C$^{F100A}$. All errors were calculated from triplicate measurements. Values for PKA-C$^{WT}$ are taken from *Walker et al., 2019*

• Supplementary file 3. Changes in enthalpy, entropy, free energy, and dissociation constants for PKI$_{5-24}$ binding to apo and ATPγN - saturated PKA-C$^{WT}$ and PKA-C$^{F100A}$. All errors were derived from triplicate measurements. The error for the cooperativity coefficient ($\sigma$) was propagated from the errors in $K_d$. Values for PKA-C$^{WT}$ were originally published in *Walker et al., 2019*.

• MDAR checklist

### Data availability
All the data generated or analyzed in this study are included in the manuscript and supporting files. The NMR chemical shifts and MD trajectories are deposited in the Data Repository for the University of Minnesota, https://doi.org/10.13020/8f9s-qd79.

The following dataset was generated:

| Author(s) | Year | Dataset title | Dataset URL | Database and Identifier |
| --- | --- | --- | --- | --- |
| Olivieri C, Cristina O | 2024 | The role of the αC-β4 loop in regulating cooperativity interaction in Protein Kinase A | https://doi.org/10.13020/8f9s-qd79 | Dyrad Digital respitory, 10.13020/8f9s-qd79 |

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
