## [Editor Report · eLife assessment]

This **important** study provides an example of integrating computational and experimental approaches that lead to new insights into the energy landscape of a model kinase. **Compelling** use of molecular dynamics simulations and NMR spectroscopy provide a conformational description of active and excited states of the kinase; one of which has not been captured in previously solved crystal structures. Overall, this comprehensive study expands our understanding of the architecture and allosteric features of the conserved bilobal kinase domain structure.

---

## [Referee Report · Reviewer #1 (Public Review)]

Summary:

The authors use insights into the dynamics of the PKA kinase domain, obtained by NMR experiments, to inform MD simulations that generate an energy landscape of PKA kinase domain conformational dynamics.

Strengths:

The authors integrate strong experimental data through the use of state-of-the-art MD studies and derive detailed insights into allosteric communication in PKA kinase. Comparison of wt kinase with a mutant (F100A) shows clear differences in the allosteric regulation of the two proteins. These differences can be rationalized by NMR and MD results. During the revision process, the authors have addressed the reviewers' comments adequately and have improved the accessibility of the manuscript to a wider audience.

---

## [Referee Report · Reviewer #3 (Public Review)]

Summary:

Combining several MD simulation techniques (NMR-constrained replica-exchange metadynamics, Markov State Model, and unbiased MD) the authors identified the aC-beta4 loop of PKA kinase as a switch crucially involved in PKA nucleotide/substrate binding cooperatively. They identified a previously unreported excited conformational state of PKA (ES2), this switch controls and characterized ES2 energetics with respect to the ground state. Based on translating the simulations into chemical shits and NMR characterizing of PKA WT and an aC-beta4 mutant, the author made a convincing case in arguing that the simulation-suggested excited state is indeed an excited state observed by NMR, thus giving the excited state conformational details.

Strengths:

This work incorporates extensive simulation works, new NMR data, and in vitro biochemical analysis. It stands out in its comprehensiveness, and I think it made a great case.

Weaknesses:

The manuscript is somewhat difficult to read even for kinase experts, and even harder for the layman. The difficulty partially arises from mixing the technical description of the simulations with the structural interpretation of the results, which is more intuitive, and partially arises from the assumption that readers are familiar with kinase architecture and its key elements (the aC helix, the APE motif, etc).

---

## [Author Response]

The following is the authors’ response to the original reviews.

**Reviewer #1:**
The very detailed insights gained by the authors into allosteric regulation require very specialized techniques in this study. This poses a challenge to communicate the methods, the results, and the meaning of the results to a broader audience. In some places, the authors overcome this challenge better than in others.

Following this reviewer’s suggestions, we have extensively revised the text, making the text more understandable to a broader audience.

The manuscript does not show up on BioRxiv.

The manuscript is now deposited in Biorxv (doi: 10.1101/2023.09.12.557419)

Fig3: GS-ES2 transition: the changes appear minimal in the illustration.

As suggested by this reviewer, we have re-examined the GS-ES2 transition and clearly defined the structural characteristics of the conformationally excited state 2 (ES2) state. As shown in the revised Fig.3 of the main text, the ground state (GS) features a π-π packing between the aromatic rings of F100 and Y156, as well as a cation-π stacking between R308 and F102. In the ES2 state, these above interactions are disrupted, while a new π-π packing interaction is formed between F100 and F102. We added new comments in the main text clarifying these structural interactions that characterize each state.

GS-ES1 transition: how is the K72-E91 salt bridge disrupted? How do you define the formation/disruption of a salt bridge? The current figure does not make this very clear and the K72-E91 salt bridge appears to be intact in ES1. Maybe the authors could replace the dotted K72-E91 line with a dotted line and distance?

As stated above, we revised Fig. 3 highlighting the differences between the two states. The K72 and E91 salt bridge is formed when the distance between Nε of K72 and Oε of E91 is shorter than 4.0 Å(the typical cutoff for a salt bridge). In the ES1 state, the outward movement of the αC helix increases the distance over 4.5 Å, disrupting the salt bridge.

L251: Could the authors remind the reader why they are only comparing V104 and I150? Could they give a little context as to why they consider the agreement to be good? It appears that they would be statistically different, so a little context for what comprises a good agreement in the literature may be helpful.

Our mutagenesis studies show that V104 and I150 are key residues for allosteric communication, and if mutated, result in well-folded but inactive kinases (Sci Adv. doi: 10.1126/sciadv.1600663). Importantly, V104 and I150 show two distinct populations in the CEST experiments that can be directly related to the GS and ES states. Regarding the fitting of these residues, we obtained a good agreement with the direction of the chemical shifts, which supports the hypothesized GS -> ES structural transition. The lack of a quantitative agreement between the chemical shifts of the experimental and simulated excited state is not surprising for two reasons (a) all state-of-the art simulations fall short in sampling slow conformational interconversions, and (b) the uncertainty of the SHIFTX algorithm for the prediction of 13C chemical shifts of methyl groups is quite large. Finally, we would like to point out that most NMR relaxation-dispersion experiments (CEST and CPMG) are performed for the backbone 15N, 13Calpha and 1H resonances, which have been used to calculate the structures of the intermediate states (Neudecker, P. et. al Science, 2012, 336,doi: 10.1126/science.1214203) and yield reasonable agreement with the prediction for metastable states derived from Markov Models (Olsson, S. J. Am. Chem. Soc., 2017,139,doi:10.1021/jacs.6b09460). To the best of our knowledge, there is no literature reporting on calculations of the 13C CEST profiles for methyl groups from MD simulations, and remarkably, we found a reasonably good agreement between experimental and predicted chemical shifts (see Fig.5C).

Just to clarify: the calculated CS values are informed by experimental CS values that were used in the calculation?

We used the backbone chemical shifts as the restraints only in the metadynamics simulations. We used the chemical shifts of the methyl groups and their corresponding excited states to verify the ES2 state.

Figure 8: in its current form this potentially exciting result is lost on the average reader.

we modified Fig. 8 of the main text, making the intra- and inter-residue correlations visible to the reader.

**Reviewer #2:**
While the alphaC-beta4 loop is a conserved feature of protein kinases, the residues within this loop vary across various kinase families and groups, enabling group and family-specific control of activity through cis and trans acting elements. F102 in PKA interacts with co-conserved residues in the C-tail, which has been proposed to function as a cis regulatory element. The authors should elaborate on the conformational changes in the C-tail, particularly in the arginine that packs against F102, in the results and discussion. This would further extend the impact and scope of the manuscript, which is currently confined to PKA.

As suggested by this reviewer, we re-analyzed the time-dependent interactions between F102 and R308 at the C-tail. As this reviewer suspected, these interactions differentiate the ES2 from the GS state. In the GS state, there is a stable cation-π interaction between F102 and R308, which becomes transient in the ES2 state (Fig. 3). For the F100A mutant, the interactions between F102 and R308 have lower occurrence relative to the WT enzyme, i.e., a weaker interaction between the αC-β4 loop and the C-tail (see new Figure 6 - figure supplement 1). The latter supports our conclusion that the structural coupling between the C-tail and the two lobes of the enzyme decreases for the F100A mutant. We added more comments in the main text.

FAIR standards of making the data accessible and reproducible are not directly addressed.

We have deposited all our NMR data on the Data Repository Site at the University of Minnesota, DRUM (https://hdl.handle.net/11299/261043).

The MD data and conformational states would be a valuable resource for the community and should be shared via some open-source repositories.

Due to the large size of the simulations (>500 GB), we could not deposit them in the DataRepository Site at the University of Minnesota (DRUM). We are actively working with the personnel at DRUM to upload all the trajectories in an alternate site. However, these data will be available to the public immediately upon request.

The authors state that ES1 and ES2 states are novel and not observed in previous crystal structures. The authors should quantify this through comparisons with PKA inactive states and with other AGC kinases.

We apologize for the confusion. We now clarify that the ES1 is a well-known inactivation pathway.As suggested by this reviewer, we now report a few examples of active and inactive conformations of PKA-C and other kinases (see new Figure 3 – figure supplement 2.). Briefly, ES1 corresponds to the typical αC-out conformation found for PKA-C bound to inhibitors or in R194A mutant. A similar conformation is present for Src, Abl, and CDK2. The αC-out conformation features a disrupted β3K-αCE salt bridge, which is key for active kinases. In contrast, the transition GS-ES2 is not present in the inactive conformations deposited in the PDB.

Based on the results, can the authors speculate on the impact of oncogenic mutations in the alphaCbeta4 loop mutations in PKA?

We now include additional comments and another citation that further supports our findings. In short, the activation of a kinase is generated by mutation insertions that stabilize the αC-β4 loop as pointed out by Kannan and Zhang (see references 28, 30, and 68). In contrast, mutations that destabilize this allosteric site (e.g., F100A) are inactivating, disrupting the structural couplings of the two lobes (our work).

**Reviewer #3:**
The manuscript is somewhat difficult to read even for kinase experts, and even harder for the layman. The difficulty partially arises from mixing technical description of the simulations with structural interpretation of the results, which is more intuitive, and partially arises from the assumption that readers are familiar with kinase architecture and its key elements (the aC helix, the APE motif, etc).

We revised the text and modified Fig. 1 in the main text to make the paper more accessible to the general audience.

The authors haven't done a good job describing the ES2 state intuitively. From my examination of the figures, it appears that in the ES2 state, the kinase domain is more elongated and the N and the C lobes are relatively less engaged than in the ground state. This may or may not be exactly, but a more intuitive description of the ES2 state is needed.

As suggested by this reviewer, we include a better description of the ES2 state of the kinase and the structural details of the inactivation pathway. Also, we checked the radius of gyration of the two lobes for GS and ES2. ES2 is slightly more elongated with an Rg of 20.3 ± 0.1 Å as compared to the GS state (20.0 ± 0.2 Å). This marginal difference is consistent with our characterization of the local packing around the αC-β4 loop, in which the lack of stable interaction with αE and C-tail in the ES2 state makes the overall structure less compact.

The authors need to introduce and give a brief description of technical terms such as CV (collective variable), PC (principal component) etc.

We now specify both collective variables and principal components and include those definitions in the Method section. Briefly, to characterize the complex conformational transitions of PKA-C, we utilize collective variables (Figure 2 – figure supplement 1). We chose these variables based on structural motifs described in the literature to define local and global structural transitions (Camilloni C., Vendruscolo, M, Biochemistry, 2015,54,7470; Kukic, P. et al. Structure, 2015,23, 745). On the other hand, we utilized the principal component analysis to compare the conformational changes of the kinase in the same two-dimensional space, revealing the two lowest frequencies that define the global motions of the enzyme (Figures 7C, D, and E).

The following paper should be discussed as it discussed similar ATP/substrate binding of Src kinase based on an extensive network that largely overlaps with the discussed PKA network. Foda, et al. "A dynamically coupled allosteric network underlies binding cooperativity in Src kinase." Nature communications 6.1 (2015): 5939.

We apologize for missing this citation. Indeed, it makes our finding more general as allosteric cooperativity is key in other kinases such as Src and ERK2. We included this in the Discussion section.

The CHESCA analysis appears to be an add-on that doesn't add much value. It is difficult to direct. I'd suggest considering removing it to the SI.

We understand this concern. We rewrote part of the paper to make the NMR analysis of the correlated chemical shifts described by the CHESCA matrices linked to the MD calculations.